# DAWP: A framework for global observation forecasting via Data Assimilation and Weather Prediction in satellite observation space

**Junchao Gong** *
Shanghai Jiao Tong University
gjchimself@sjtu.edu.cn

**Jingyi Xu** *
Fudan University
jyxu22@m.fudan.edu.cn

**Ben Fei** †
The Chinese University of Hong Kong
Shanghai AI Laboratory
benfei@cuhk.edu.hk

**Fenghua Ling**
Shanghai AI Laboratory
lingfenghua@pjlab.org.cn

**Wenlong Zhang**
Shanghai AI Laboratory
zhangwenlong@pjlab.org.cn

**Kun Chen**
Shanghai AI Laboratory
chenkun@pjlab.org.cn

**Wanghan Xu**
Shanghai AI Laboratory
xuwanghan@pjlab.org.cn

**Weidong Yang**
Fudan University
wdyang@fudan.edu.cn

**Xiaokang Yang**
Shanghai Jiao Tong University
xkyang@sjtu.edu.cn

**LEI BAI** †
Shanghai AI Laboratory
bailei@pjlab.org.cn

## Abstract

Weather prediction is a critical task for human society, where impressive progress has been made by training artificial intelligence weather prediction (AIWP) methods with reanalysis data. However, reliance on reanalysis data limits the AIWPs with shortcomings, including data assimilation biases and temporal discrepancies. To liberate AIWPs from the reanalysis data, observation forecasting emerges as a transformative paradigm for weather prediction. One of the key challenges in observation forecasting is learning spatiotemporal dynamics across disparate measurement systems with irregular high-resolution observation data, which constrains the design and prediction of AIWPs. To this end, we propose our DAWP as an innovative framework to enable AIWPs to operate in a complete observation space by initialization with an artificial intelligence data assimilation (AIDA) module. Specifically, our AIDA module applies a mask multi-modality autoencoder (MMAE) for assimilating irregular satellite observation tokens encoded by mask ViT-VAEs. For AIWP, we introduce a spatiotemporal decoupling transformer with cross-regional boundary conditioning (CBC), learning the dynamics in observation space, to enable sub-image-based global observation forecasting. Comprehensive experiments demonstrate that AIDA initialization significantly improves the rollout and efficiency of AIWP. Additionally, we show that DAWP holds promising potential to be applied in global precipitation forecasting. Code will be available at this github repo.

*Equal Contribution

†Corresponding Authors: Ben Fei (benfei@cuhk.edu.hk) and Lei Bai (bailei@pjlab.org.cn)

39th Conference on Neural Information Processing Systems (NeurIPS 2025).

# 1 Introduction

Weather prediction is a critical task that significantly impacts various socioeconomic aspects, including transportation, agriculture, and public safety. Traditional numerical weather prediction (NWP) systems rely on intricate human-designed workflows [1, 2], such as numerical assimilation systems and physical solvers, to generate global precipitation predictions.

Recently, transformative progress has been made by artificial intelligence weather prediction (AIWP) models. These AIWP models now achieve forecast skill scores comparable to or even surpassing those of leading physics-based NWP systems [3, 4, 5, 6]. To learn the atmospheric dynamics, reanalysis products are widely used [7, 8, 9, 10].

However, reanalysis data, generated by numerical data assimilation, introduce intrinsic limitations in AIWP models built upon them. (I) **Data Assimilation Biases:** (Re)analysis products are synthesized by numerical data assimilation (DA), where direct observations are blended with a physics-based forecast. During DA, information loss of direct observation occurs due to the limited utilization of raw observational data and the preprocessing that resamples observations to regular grids of reanalysis data format with finite resolution [11, 12, 13, 14]. Additionally, the incomplete physical process modeling and uncertainty parameterizations in physics-based forecast systems also introduce biases, which could hinder learning the actual dynamics of the atmosphere [15]. (II) **Temporal Discrepancies:** The temporal lag between direct observation acquisition (nearly real-time) and analysis data generation (up to six hours) severely degrades the quick response ability of AIWP models [2, 16]. These limitations may be further exacerbated by the discrepancy between real-world observation space and physical forecasting space which is required by NWP systems to implement dynamic equations of atmospheric [17]. As AIWP models do not require physical solvers to predict the evolution of the atmosphere, there is potential for them to directly predict atmosphere states in real-world observation space.

Artificial Intelligence Direct Observation Prediction (AI-DOP) is emerging as a transformative, data-driven approach with the potential to overcome the limitations in reanalysis-driven AIWP methods. The key challenge of AI-DOP is learning the spatiotemporal dynamics not only within a single observation source but also across disparate measurement systems, given the irregular and high-resolution observation data [17]. The spatiotemporal modeling approaches are restricted to learn the relationships between different observations with irregular data. [17] applies a transformer with mask tokens to reformat multiple observations into regular ones. Further, [18] uses a graph encoder to flexibly encode different measurements into a uniform latent representation for latent forecasting with a naive transformer backbone. In addition, as AI-DOP requires being generalized to any location grids given observations with variable missing values [17], the dense forecasts and sparse observation inputs result in **input-output distribution shift** in rollout as shown in (b) and (c) of Figure 1. Motivated by these questions, we argue that training AIWP models in **a uniform observation space** where input and output are both regular grid data.

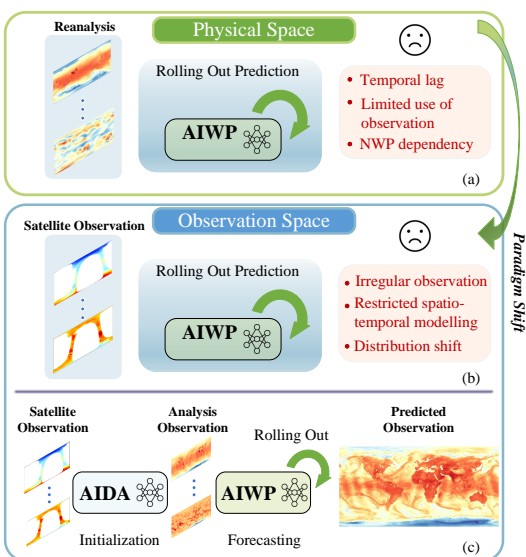

Figure 1: A paradigm shift from physical space to observation space. Our DAWP is illustrated in (c).

We propose our DAWP, an AI-DOP system composed of an observation space data assimilation (AIDA) module and an AIWP module, to learn the spatiotemporal dynamics between various satellite observations with irregular and high-resolution characteristics. To begin with, a transformer VAE encoder/decoder is designed with observation masks to regionally encode high-resolution irregular direct earth observations for efficient I/O and computation. Then, we implement a sub-image-based AIDA module for observation space data assimilation by a multi-modal masked autoencoder with en-

coded observation tokens. By learning the spatiotemporal correlations between various observations in the AIDA initialization, irregular observations are transformed into a uniform completed observation space. In this imputed space, we train our sub-image-based AIWP module with Cross-regional Boundary Conditioning (CBC), which could forecast observations with a global state cache providing atmospheric states of neighbours. Finally, a combination of mapping operators is used to obtain global observation predictions or precipitation variables. We conduct comprehensive experiments to demonstrate the effectiveness of our DAWP framework for global direct observation predictions. We summarize the contributions of this paper as follows:

- **Innovative framework** integrating AIDA and AIWP methods for direct observation predictions: We propose a brand-new framework that leverages an observation space AIDA module, transforming irregular observations into a uniform observation space, with AIWP modules to achieve skillful direct observation predictions, bypassing the limitations of reanalysis data.
- **High-resolution global forecasting** for observation and precipitation: We introduce a mask ViT-VAE and a spatiotemporal transformer with cross-regional boundary information for encoding high-resolution irregular observations, and implement global forecasting efficiently on sub-images of global observations.
- **Comprehensive experiments**: We organize a composite satellite observation dataset with a size of over 35TB, which has a spatiotemporal resolution of $12\times1152\times2304$. Comprehensive experiments and reanalyses are presented, demonstrating the effectiveness of our DAWP framework and the potential of direct observation predictions.

## 2    Related work

**Weather prediction with deep learning.** Recent studies have demonstrated that machine learning systems can produce accurate medium-range forecasts, comparable to physics-based models, for key weather parameters [3, 4, 6, 5, 19, 20, 21, 22]. FourcastNet was the first to propose using deep neural networks to learn global atmospheric dynamics [23]. By scaling up the training stage, Pangu-Weather [3] and GraphCast [4] simultaneously achieved accuracy levels comparable to those obtained by the operational IFS systems at ECMWF. Other works extend the AIWP from aspects including forecasting skill [5, 6], resolution [19], probabilistic modelling [24], and physics informed [25]. Although impressive progress has been made, previous AIWP methods still rely on reanalysis data, which introduces inherent limitations, including temporal lag, limited observation use, and dependency on NWP systems.

**Direct observation prediction.**    Direct observation prediction holds transformative potential for overcoming the dependency on reanalysis, enabling the forecasting of weather using direct observations. Although the concept of direct observation prediction has been widely applied in fields such as precipitation nowcasting, where radar echoes are utilized for short-term forecasting [26, 27, 28, 29, 30], its application in global weather prediction remains limited. In contrast to gridded radar observations, direct observations of the global atmosphere are irregular and non-gridded, making global weather DOP challenging. Transformer-DOP proposes using a transformer with mask tokens to handle the irregular global Earth observations [17]. Simultaneously, Graph-DOP employs graph neural networks to flexibly encode direct observations [18]. EarthNet proposes pretraining the backbone with an observation assimilation task and then finetuning it with a prediction task [31]. Although they successfully apply irregular global observation for prediction, these designs suffer from rollout distribution and ineffective spatiotemporal learning as the input space is discrete while the output space is dense. We propose applying AIDA to transform the input space into a dense one, thereby solving the misalignment between input and output.

## 3    Method

Our DAWP is designed for global observation forecasting in a uniform satellite observation space. The key components of DAWP are an observation space data assimilation module and a cross-regional boundary conditioning weather prediction module. Additionally, we introduce the mask ViT-VAE to encode observations and produce precipitation variables. Our DAWP is illustrated in Figure 2.

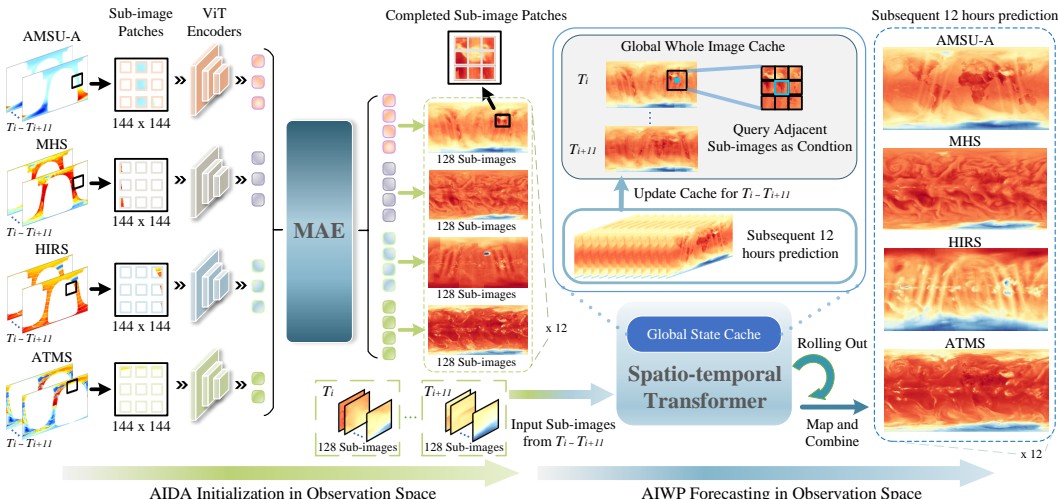

Figure 2: The framework of our DAWP. There are two stages in our DAWP: (1) Initialization and (2) Forecasting.

## 3.1 Initialization: observation space assimilation by multi-modal masked autoencoder

The missing and sparse observations, attributed to the inherent characteristics of orbital motion, present an irregular input observation space, as shown in the left column of Figure 2. Directly taking these satellite observations as inputs not only restricts the network design for spatiotemporal modeling but also leads to a distribution shift when implementing rollout forecasting, as the output space is required to be a regular one. To meet the gap, we propose using observation space assimilation with a Multi-modal Masked Autoencoder [32, 33] (MMAE) as the initialization stage for direct observation predictions. The MMAE fills in missing areas by leveraging contextual information from different sensors and spatiotemporally nearby observations.

The core of our assimilation module is a naive MAE that imputes masked multi-modal satellite tokens following [34, 32, 31]. Since imputation mainly relies on space-time nearby observations from multiple satellite observations, our MMAE sub-regionally processes data from multiple sources within a fixed time window of 12 and a sub-image of 144×144. Observations in the time window are tokenized frame by frame by pretrained satellite-specific mask ViT encoders, where missing patches are ignored. Remaining spatiotemporal tokens from each satellite are concatenated and passed through MAE for complete missing information in the observation space. For MAE training, we randomly mask a given number of tokens from the whole concatenated remaining tokens and reconstruct the left observed but masked tokens. As the number of observed tokens from different time windows can vary, we flexibly pad [EOS] tokens to maintain a uniform sequence length for efficient attention computation. In the inference stage, masks for MAE are released to utilize as many available observations as possible.

## 3.2 Forecasting: cross-regional boundary conditioning direct observation prediction

After AIDA, we implement an efficient weather prediction module for direct observation prediction through cross-regional boundary conditioning in the imputed observation space, initialized by our assimilation module. For efficiently integrating with the assimilation module pretrained by sub-images, our weather prediction module is also applied to sub-images generated by the assimilation module. Since sub-images only contain local atmospheric states, predictions on sub-images require cross-regional information interaction for continuous spatiotemporal modeling. We introduce a global state cache to store observation states for cross-regional boundary conditioning during forecasting.

With the global state cache, our weather prediction module achieves efficient cross-regional boundary conditioning observation forecasting by applying spatiotemporal decoupling attention structure. In the forecasting stage, the assimilated observations simplify the prediction task into a standardized spatiotemporal forecasting problem, which is widely explored [35, 36, 37]. We follow the concept of spatiotemporal decoupling in spatiotemporal forecasting [35, 30]. Specifically, our weather prediction

Table 1: Dataset overview.

| Modality/Sensor | Satellite | Channels | Level | Period |
|---|---|---|---|---|
| Advanced Microwave Sounding Unit-A (AMSU-A) [40] | NOAA18 & 19 | Microwave radiance 15 bands | 1B | 2007-2023 |
| Advanced Technology Microwave Sounder (ATMS) [41, 42] | NPP & NOAA20 | Brightness temperature 9 bands | 1C | 2012-2023 |
| High Resolution Infrared Radiation Sounder (HIRS) [40] | NOAA18 & 19 | Infrared radiance 20 bands | 1B | 2007-2023 |
| Microwave Humidity Sounder (MHS) [40] | NOAA18 & 19 | Microwave radiance 5 bands | 1B | 2007-2023 |
| ATMS-Precipitation [43, 44] | NPP & NOAA20 | Precipitation product 2 channels | 2A | 2012-2023 |

module is composed of $N$ Temporal-Spatial (TS) attention blocks [37, 36]. Attention blocks have the advantages of simplicity and scalability, while the TS spatiotemporal decoupling reduces the sequence length of spatiotemporal tokens for efficient computation. Utilizing the efficiency of TS decoupling, we simply pad tokens from neighbouring areas to the tokens of the prediction region as SD3 [38] and Flux [39] do. In this way, cross-regional boundary information of the border area is passed to the center forecasting region. To forecast multistep global observations, we maintain a global state cache during the inference phase, ensuring consistent updates for subsequent steps. Specifically, when forecasting observations of center sub-images, current adjacent sub-images are queried as conditions from the state cache according to the relative coordinates. After one step of global prediction has been completed, the global state cache is updated with the subsequent 12 hours prediction to support rollout forecasting. More details can be found in the Appendix A.

## 3.3 Encoding and precipitation mapping via mask ViT-VAE

In our DAWP, we introduce a mask ViT-VAE both for encoding and mapping multiple-channel satellite observations with missing values. The mask ViT-VAE consists of a vision transformer (ViT) encoder/decoder with masks enabling the model to ignore patches without sufficient observations. The ViT encoder/decoder provides better compression capability, as detailed in Appendix D, compared to SD-VAE for satellite observations, which typically have more channels than natural images. Moreover, it explicitly maintains spatial consistency between tokens and pixels by position embeddings and mitigates the influence of missing values through mask attention. With our mask Vit-VAE, encoding/decoding is pretrained as a reconstruction task, while the precipitation mapping is trained with ATMS inputs and ATMS-precipitation outputs.

## 4 Experiments

In the experiment part, a comprehensive analysis of our DAWP is presented. First, we introduce the composition of our data and training details. Based on the observation data, a comparison of our DAWP with other AI-DOP methods is implemented. Further, we evaluate the precipitation forecasting skill of these AI-DOP methods by applying a precipitation mapping network. In addition to evaluating DAWP's capabilities of forecasting capabilities, we also conducted ablation studies to validate the effectiveness of our modular designs. Finally, the importance of each satellite for DOP is tested by ablating the input modalities of the assimilation module.

### 4.1 Experimental Setups

**Data.** The top four datasets listed in Table 1 are used for training our DAWP, while ATMS-precipitation is used to train the precipitation mapping with ATMS. We generate hourly, $0.16°$ resolution composites by interpolating and reprojecting the raw data as detailed in Appendix E. Additional information for dataset split and introduction is presented in Appendix F.

**Training details.** The training details are presented in Appendix G. Training our mask ViT-VAE, MMAE AIDA, and TS decoupling AIWP takes about 1 day, 5 days, and 4 days, respectively. Besides, our DAWP is compared with the persistence model, using our AIDA module for completion of missing values, as mentioned in [35]. We replicate the EarthNet [31] and Transformer-DOP [26] models on our composite dataset, as their codes and data are closed source.

Table 2: MAE error of forecasting during 3 lead time periods (0-12h, 12-24h, and 24-36h) for different channels of the satellite data. We use the unit of 1e-5 for AMSU-A, 1e-4 for MHS, and 1e-0 for both ATMS and HIRS.

| Methods | Lead time | AMSU-A | | ATMS | | HIRS | | MHS | |
|---|---|---|---|---|---|---|---|---|---|
| | | ch0 | ch1 | ch0 | ch1 | ch9 | ch10 | ch0 | ch1 |
| Persistence [35] | 0-12h | 5.86 | 9.15 | 14.37 | 11.69 | 12.43 | 2.21 | 7.01 | 14.74 |
| EarthNet [31] | | 2.93 | 4.89 | 6.96 | 6.14 | 9.15 | 1.39 | 3.96 | 9.65 |
| Transformer-DOP [17] | | 2.67 | 4.48 | 6.40 | 5.61 | 9.22 | 1.42 | 3.91 | 9.48 |
| Ours | | 1.92 | 3.39 | 3.36 | 3.27 | 7.70 | 1.12 | 3.07 | 7.91 |
| Persistence [35] | 12-24h | 4.35 | 6.94 | 10.40 | 8.86 | 13.58 | 2.30 | 6.07 | 15.09 |
| EarthNet [31] | | 4.12 | 6.65 | 11.25 | 9.00 | 11.14 | 1.98 | 5.46 | 13.11 |
| Transformer-DOP [17] | | 3.84 | 6.14 | 10.04 | 8.04 | 11.08 | 1.95 | 5.19 | 12.65 |
| Ours | | 3.11 | 5.12 | 7.35 | 6.35 | 9.57 | 1.54 | 4.51 | 10.54 |
| Persistence [35] | 24-36h | 6.39 | 9.84 | 15.37 | 12.52 | 14.61 | 2.61 | 8.00 | 17.86 |
| EarthNet [31] | | 5.17 | 8.14 | 12.52 | 10.08 | 12.37 | 2.36 | 6.41 | 15.08 |
| Transformer-DOP [17] | | 4.91 | 7.54 | 11.35 | 9.07 | 12.39 | 2.27 | 6.22 | 14.70 |
| Ours | | 3.66 | 5.80 | 7.84 | 6.81 | 10.71 | 1.79 | 5.15 | 12.22 |

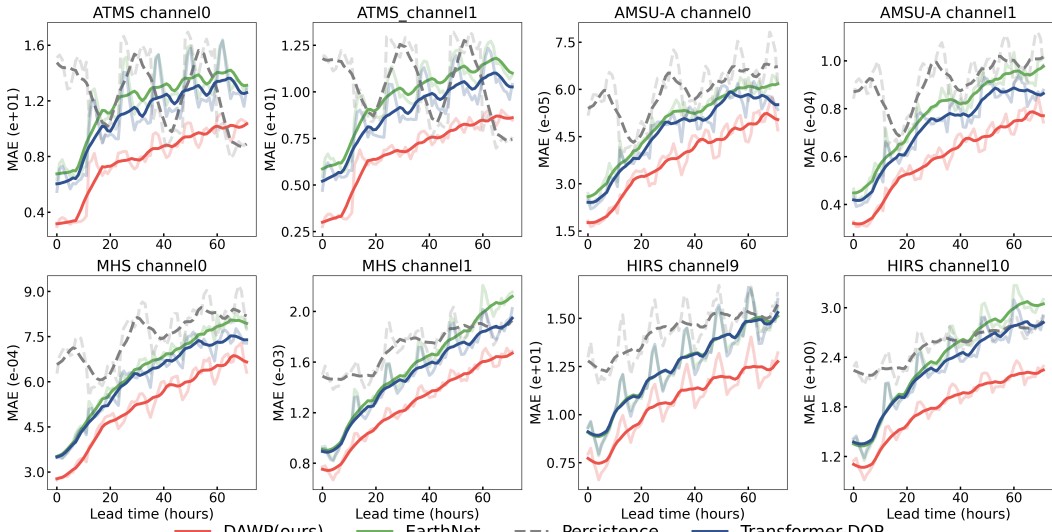

Figure 3: Curves of MAE for the prediction of different modalities. The max leadtime is 72h with a 1h temporal resolution.

## 4.2 Direct observation prediction

We compare our DAWP with other AI-DOP methods to evaluate the ability to predict direct observations. The metric for evaluation is the mean absolute error (MAE) between the predictions and observed values. The baselines are also trained with sub-images as our DAWP. Among them, persistence is a naive method that uses the last observation as the prediction of the next one.

The time-averaged MAEs within fixed time windows of 0-12h, 12-24h, and 24-36h are presented in Table 2. We select two channels with meaningful patterns of each modality observation to calculate the MAE. The results show that our DAWP significantly outperforms other methods. Specifically, our DAPW's MAE on AMSU-A among 24-36h is 3.66 and 5.80, which not only outperforms those of EarthNet and Transformer-DOP, but also surpasses EarthNet's (3.84 and 6.14) and Transformer-DOP's (4.12 and 6.65) MAEs on AMSU-A among 12-24h. It is the same when comparing our DAWP with baselines on other modalities. These results indicate our DAWP has a 12-hour lead time advantage in direct observation prediction. To provide a more detailed analysis of the observation prediction, we present the MAE figure in a time range of 0-72h in Figure 3. It can be observed that the 1h temporal resolution of the figure depicts the periodicity of the prediction errors. This periodicity is attributed to the strip-scanning characteristics of polar-orbiting satellites, which generate periodic

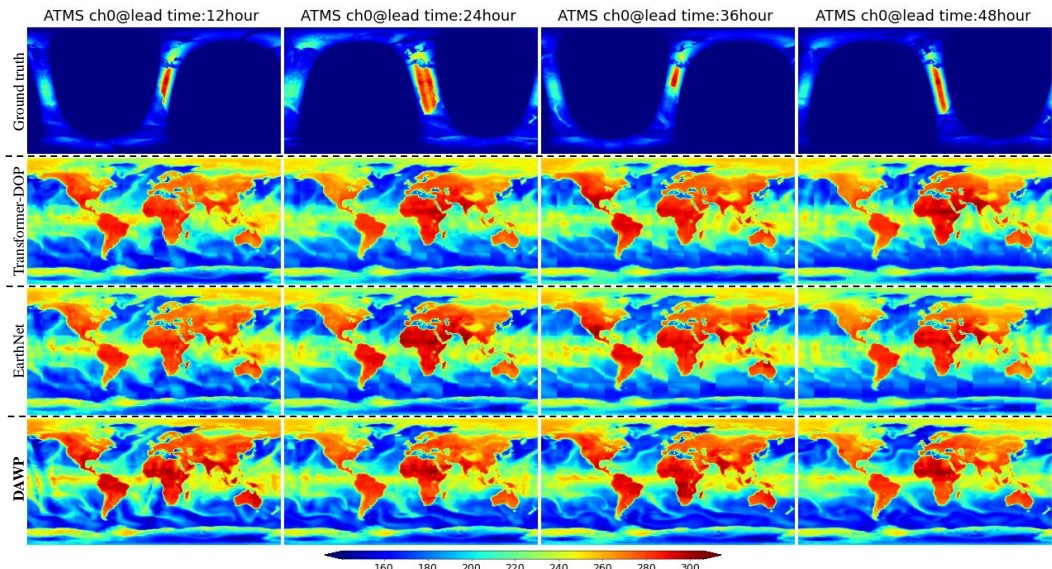

ATMS ch0@lead time:12hour  ATMS ch0@lead time:24hour  ATMS ch0@lead time:36hour  ATMS ch0@lead time:48hour

Figure 4: A visualization of rollout predictions for global satellite observation forecasting.

Table 3: Forecasting skills in 12 hours on precipitation-related variables Total Column Water Vapor (TCWV) and Surface Precipitation (SP). CSI and FAR scores are calculated on different thresholds.

| Method | TCWV (mm) | | | | | | SP (mm/h) | | | | | |
|---|---|---|---|---|---|---|---|---|---|---|---|---|
| | CSI-10 | CSI-20 | CSI-30 | FAR-10 | FAR-20 | FAR-30 | CSI-0.5 | CSI-1.0 | CSI-2.0 | FAR-0.5 | FAR-1.0 | FAR-2.0 |
| Persistence [35] | 0.853 | 0.702 | 0.636 | 0.121 | 0.266 | 0.332 | 0.110 | 0.073 | 0.031 | 0.844 | 0.861 | 0.655 |
| EarthNet [31] | 0.909 | 0.822 | 0.786 | 0.047 | 0.130 | 0.172 | 0.127 | 0.050 | 0.008 | 0.666 | 0.692 | **0.180** |
| Transformer-DOP [17] | 0.905 | 0.822 | 0.789 | 0.053 | 0.130 | 0.169 | 0.136 | 0.057 | 0.010 | 0.655 | 0.685 | 0.214 |
| Ours | **0.917** | **0.841** | **0.807** | **0.034** | **0.088** | **0.120** | **0.197** | **0.102** | **0.035** | **0.529** | **0.577** | 0.283 |

observations. Although there is periodicity in the MAE, our DAWP consistently outperforms other methods, establishing a lead time advantage from the beginning. Specifically, in subfigure (a), at the beginning of the prediction, our DAWP's MAE is about 3.5 while the MAE of EarthNet is about 6.8, which is almost 2 times larger than ours. In addition, EarthNet is surpassed by the naive persistent baseline at the lead time of 16h, but our DAWP doesn't meet the persistent baseline until about 64h.

In Figure 4, we visualize the prediction results of these methods at different lead times. At the lead time of 12 hours (the first prediction step), our DAWP exhibits a slightly better prediction than others. When the lead time increases, a significant distortion appears in the predictions of EarthNet and Transformer-DOP, while our DAWP maintains a relatively stable structure. It has the same trend as the MAE curve in Figure 3, indicating that our DAWP is more robust in rollout predictions.

The results of direct observation prediction indicate that our DAWP outperforms other methods in the 0-72h time range and selected channels, demonstrating the effectiveness of our method in both initial prediction and rollout prediction. More evaluations on other channels are shown in the Appendix I.

## 4.3 Global precipitation forecasts

In this section, we evaluate the potential of AI-DOP methods for global precipitation forecasting. A precipitation mapping network is trained to transform observation predictions into precipitation products. It is worth noting that in this way, our DAWP does not require satellite precipitation products as input, making the forecasting a quick response to observations.

The precipitation skill is evaluated by the Critical Success Index (CSI) [45] and False Alarm Ratio (FAR) [46] metrics on observed points in a time range of 0-12h. CSI measures the ability of the model to correctly identify precipitation events, while FAR assesses the reliability of the model's predictions. The combined application of CSI and FAR enables a comprehensive precipitation forecasting skill assessment. We present a detailed definition of CSI and FAR in Appendix H.

Table 3 presents the quantitative results of the precipitation forecasting skill of AI-DOP methods on total column water vapor index (TCWV) and surface precipitation (SP). The thresholds for TCWV and SP are set to [10mm, 20mm, 30mm] and [0.5mm/h, 1.0mm/h, 2.0mm/h], respectively. On variable TCWV, our DAWP achieves a slight advantage in CSI over other methods. Specifically, our DAWP's achieves 0.807 on CSI-30, while Earthnet and transformer-dop achieve 0.786 and 0.789, respectively. This advantage is maintained across all thresholds. In terms of FAR, our DAWP is significantly lower than other methods. It indicates that our DAWP predicts more accurate TCWV with lower false alarm rates than baseline methods, demonstrating the reliability of our DAWP. We also analyze the forecasting skill on SP variable. The CSI of our DAWP is significantly higher than that of other methods on CSI-0.5 and CSI-1.0, achieving 0.197 and 0.102. Compared with transformer-dop, our DAWP increases CSI-0.5 and CSI-1.0 by 44.8% and 78.9%. Another observation about CSI is a decreasing trend with increasing thresholds. Especially, when the threshold increases to 2.0, the CSI of EarthNet and transformer-dop are even lower than that of the naive persistent model. Only our DAWP maintains a slightly higher forecast skill on CSI 2.0. The decrease of forecasting skill is caused by the temporal decay of prediction intensity [26, 29], which could further hamper the mapping network's performance. When evaluating FAR on threshold 2.0mm/h, our DAWP uncommonly surpasses Earthnet and transformer-dop slightly. It could be explained by considering the CSI-2.0 results, as it is difficult for EarthNet and transformer-dop to produce predictions greater than 2.0mm/h, resulting in few false alarms. Besides, compared to the persistent baseline with a comparable CSI-2.0 skill, our DAWP has a FAR score which 57.8% lower than persistent's, showing stronger reliability. The evaluation of CSI and FAR scores demonstrates the potential of our DAWP for global precipitation forecasting, simultaneously increasing the accuracy and reliability for precipitation forecasting skill of AI-DOP methods.

## 4.4 Ablation study

**Effect of AIDA initialization.** An ablation study is conducted to validate the effectiveness of our AIDA module on our DAWP and other AI-DOP methods. The results demonstrate that AIDA enables efficient spatiotemporal modeling and enhances the rollout prediction ability of AI-DOP methods by imputing the observation space.

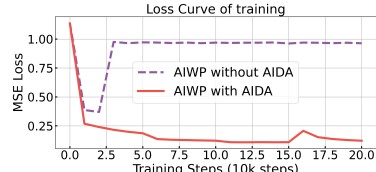

Figure 5: Training loss curve.

The training is unstable after cancelling the AIDA stage in our DAWP. We show the curve of training loss in Figure 5. It can be observed that the training loss of our DAWP with AIDA is decreasing steadily, while the loss curve without the initialization of AIDA dramatically increases at about 20k training steps and maintains a MSE loss of 1.0. In contrast, the loss of our DAWP with AIDA could converge to less than 0.1 after 200k training steps. This phenomenon indicates that the classical spatiotemporal learning methods are difficult to learn the dynamics in an observation space with variable missing values. By imputing the observation space into a completed spatiotemporal space, AIDA benefits efficient spatiotemporal learning modeling.

Applying AIDA initialization enhances the rollout prediction ability of AI-DOP methods. The quantitative results of transformer-dop are depicted in Figure 6. We exhibit the averaged MAE of 0-72h prediction on each sensor with-

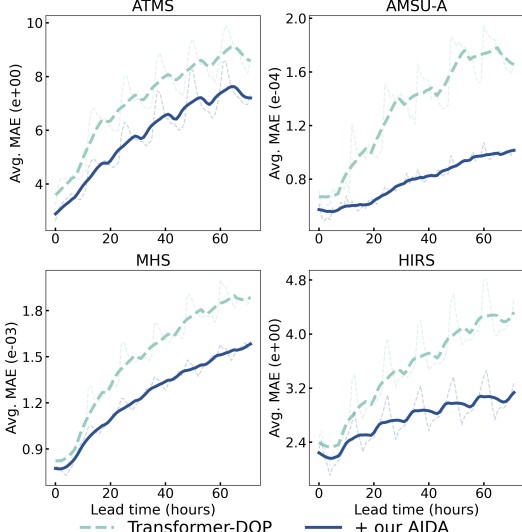

Figure 6: The MAE curves of Transformer-DOP with(w/o) our AIDA initialization.

/without AIDA initialization. Transformer-DOP with AIDA initialization significantly outperforms the original Transformer-DOP when the lead time increases. It reveals the potential of AIDA initialization for boosting the AI-DOP model for multi-step rollout prediction.

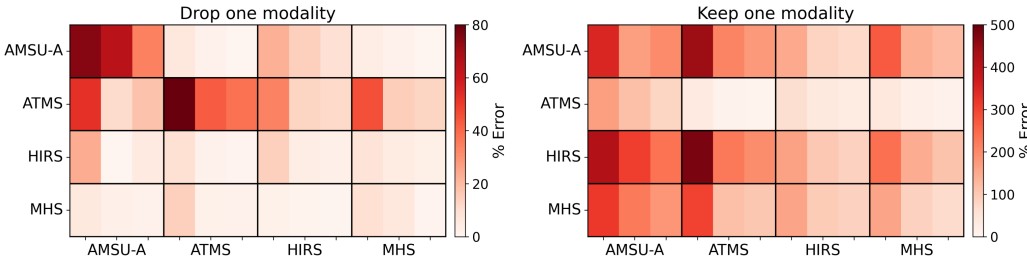

Figure 8: Matrix of relative errors under the setting of dropping one modality and keeping one modality. The three columns in a black rectangle represent the relative MAE error ratios of 0-12h, 12-24h, and 24-36h lead times, respectively.

**Gains of cross-regional boundary conditioning.** We compare the observation predictions with and without cross-regional conditions by training a DAWP without neighbour regions as inputs. This ablation study is conducted to verify the effect on the accuracy and spatial continuity of the prediction.

The convergence loss of our DAWP with and without cross-regional condition is presented in Table 4, showing conditioning on neighbour regions improves the prediction accuracy. In this table, we evaluate the convergence loss of center areas and neighbor areas. For DAWP without Cross-regional Boundary Conditioning (CBC), we take the center area as the neighbour area. The average convergence loss of the center area is 0.106, which is significantly than that of the neighbour area. Besides, it also significantly outperforms the DAWP without CBC on the

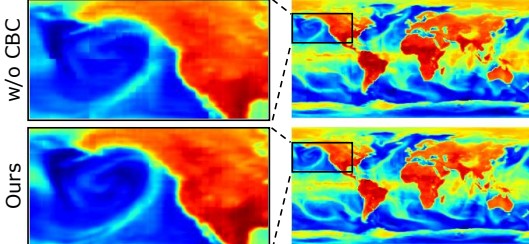

Figure 7: A visualization of forecasting results with(w/o) Cross-regional Boundary Conditioning.

centre area by about 15.6%. This result validates the necessity of cross-regional conditions for weather prediction, as the boundary information of atmospheric physical motion is crucial for accurate weather forecasting.

Another benefit of cross-regional conditions is the improvement of prediction continuity, as shown in Figure 7. The first and second rows show the predictions of ATMS channel 0 at a lead time of 6 hours without and with CBC, respectively. It can be observed in the black box that the continuity of the adjacent regions is improved, which is helpful for keeping the atmospheric structure.

**Modality sensitivity analysis.** We introduce the experiment of using different modality combinations for multistep observation prediction to gain insight into the importance of each modality's data. As shown in Figure 8, a drop one and a keep one combination are explored. The setting of drop one means removing one modality's observation before AIDA initialization and con-

Table 4: Converged loss of our DAWP with and without CBC module. The average loss is calculated by averaging the loss of all four modalities.

| Module | Area | Modalitiy | | | | Avg. |
|---|---|---|---|---|---|---|
| | | AMSU-A | ATMS | HIRS | MHS | |
| w/o CBC | border | 0.063 | 0.101 | 0.236 | 0.111 | 0.128 |
| | center | 0.063 | 0.101 | 0.236 | 0.111 | 0.128 |
| with CBC | border | 0.069 | 0.098 | 0.324 | 0.101 | 0.148 |
| | center | **0.054** | **0.074** | **0.214** | **0.084** | **0.106** |

ducting multi-step prediction, while in the keep one setting, we only keep one modality's observation before AIDA initialization for forecasting. We evaluate the influence of each modality by calculating the relative MAE error ratios between the MAE of DAWP with completed modality inputs.

The result of drop one is shown in the left of Figure 8. Row names of the figure indicate the modality that is dropped, while columns sequentially included in a modality name represent the relative MAE error ratio of this modality in time windows 0-12h, 12-24h, and 24-36h. The overall trend of the drop one setting demonstrates that dropping any modality's data leads to an increase of MAE error. Besides, it's observed that for the rollout predictions, the MAE error ratio is gradually decreasing, indicating the insufficient usage of observations for multistep predictions. When focusing on single modalities, we find that HIRS and MHS can still achieve a relatively low MAE error ratio when their own data is dropped, indicating that they have redundant information for the prediction.

The error ratio matrix of keep one setting is also shown in Figure 8. When only keeping ATMS observation, other modalities' MAE error ratios of predictions are the lowest, which are even lower than keeping the satellites themselves. This indicates that ATMS has a substantial amount of information for prediction. In contrast, for predicting one modality itself, AMSU-A exhibits the highest MAE error ratio, showing that it has the least spatiotemporal dynamics information.

## 5   Conclusion

In this paper, we propose DAWP, a novel framework using AIWP for observation prediction with an AIDA module as initialization. Comprehensive experiments are conducted to validate the efficiency and potential of our DAWP framework for observation forecasting and downstream applications such as precipitation forecasting. **Broader Impacts&Future Work**: First, our framework readily integrates variable observations, demonstrating its potential as an implicit Earth system modeling framework. Second, our framework has broad application prospects. It can seamlessly adapt to diverse downstream tasks-such as surface parameter estimation, wildfire monitoring, and sea ice mapping-whenever observations or retrieval operators are available, similar to precipitation forecasting. Third, our framework holds a promising potential for directly predicting physical variables by integrating observations of weather variables such as station data. **Limitations**: Although our DAWP framework improves the observation forecasting, the sources of observation are still homogeneous in satellite observations. More observation sources will be integrated with DAWP in the future.

## Acknowledgements

This work is supported by Shanghai Artificial Intelligence Laboratory and the JC STEM Lab of AI for Science and Engineering, funded by The Hong Kong Jockey Club Charities Trust, the Research Grants Council of Hong Kong (Project No. CUHK14213224). This work was done during Junchao Gong's internship at Shanghai Artificial Intelligence Laboratory.

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

# A  Global state cache

In this section, we clarify the design of the global state cache. It is composed of a current cache and a previous cache. The current one is used to store the predicted sub-images of $T_{12i:12(i+1)}$, and the previous one, storing sub-images of $T_{12(i-1):12i}$ is used to provide neighbour information for the prediction.

```python
class GlobalStateCache:
    def __init__(self, domains, domain_Chs, time_window,):
        self.domains = domains
        self.time_window = time_window
        self.domain_Chs = domain_Chs

        self.cur_cache = self.init_cache()
        self.prev_cache = self.init_cache()

    def init_cache(self):
        domain_cache = {}
        for domain in self.domains:
            domain_Ch = self.domain_Chs[domain]
            # original image size is 1152x2304.
            # 8x16 subimages with the size of 144x144.
            # 9x9 is the number of tokens in a subimage.
            domain_cache[domain] = /
            torch.zeros((8, 16, self.time_window, domain_Ch, 9, 9))
        return domain_cache
```

The main operators of the global state cache are query neighbors and update cache. Query neighbors is used to get the adjacent sub-images as conditions for the prediction of central areas. Update cache is used to update the previous and current cache with the subsequent 12 hours predictions.

```python
    def query_neighbours(self, rel_coordinate):
        neighbour_coords = self._get_8_neighbour_coord(rel_coordinate)
        neighbours = {}
        for domain in self.domains:
            neighbours[domain] = []
            for coord in neighbour_coords:
                neighbours[domain].append(
                    self.prev_cache[domain][coord[0], coord[1]]
                )
        return neighbours

    def update_cache(self, rel_coordinate, pred_subimg):
        for domain in self.domains:
            coords = rel_coordinate[0], rel_coordinate[1]
            self.cur_cache[domain][coords] = pred_subimg[domain]
        if have_pred_whole_img:
            self.prev_cache = self.cur_cache
        return None
```

To get the neighbour coordinates from a Plane Rectangular Coordinate System, we utilized a _get_8_neighbour_coord function that incorporates Earth's spherical geometry, specifically handling the left-right and top-bottom boundaries of the image.

```python
    def _get_8_neighbour_coord(self, rel_coordinate, h=8, w=16):
        """
        h, w is the height and width of the image
        r, c is the coordinate of the pixel
        case 1: if the pixel is in the center of the image, return all
                                              8 neighbours
        case 2: if the pixel in w border, treat the image as h border
                                      is connected
```

```python
        case 3: if the pixel in h border, symmetrically get the
                                        neighour
        case 4: if the pixel in corner, use the rule of both w border
                                        and h border
    ret:
    8 neighbours ordered as [up_left, up, up_right, left, right,
                                        down_left, down, down_right
                                        ]
    """
    r, c = rel_coordinate[0], rel_coordinate[1]
    assert r >= 0 and r < h
    assert c >= 0 and c < w
    w_border_flag = (c == 0 or c == w - 1)
    h_top_border_flag = (r == 0)
    h_bottom_border_flag = (r == (h - 1))

    neighbours = []
    if not (h_top_border_flag or h_bottom_border_flag):
        up_left = ((r - 1) % h, (c - 1) % w)
        up = ((r - 1) % h, c)
        up_right = ((r - 1) % h, (c + 1) % w)
        left = (r, (c - 1) % w)
        right = (r, (c + 1) % w)
        down_left = ((r + 1) % h, (c - 1) % w)
        down = ((r + 1) % h, c)
        down_right = ((r + 1) % h, (c + 1) % w)
    elif h_top_border_flag:
        up_left = (r, (c + 1 + w//2) % w)
        up = (r, (c + w // 2)% w)
        up_right = (r, (c - 1 + w//2) % w)
        left = (r, (c - 1) % w)
        right = (r, (c + 1) % w)
        down_left = ((r + 1) % h, (c - 1) % w)
        down = ((r + 1) % h, c)
        down_right = ((r + 1) % h, (c + 1) % w)
    elif h_bottom_border_flag:
        up_left = ((r - 1) % h, (c - 1) % w)
        up = ((r - 1) % h, (c)% w)
        up_right = ((r - 1) % h, (c + 1) % w)
        left = (r, (c - 1) % w)
        right = (r, (c + 1) % w)
        down_left = (r, (c + 1 + w//2) % w)
        down = (r, (c + w//2)% w)
        down_right = (r, (c - 1 + w//2) % w)
    else:
        raise NotImplementedError

    neighbours.append(up_left)
    neighbours.append(up)
    neighbours.append(up_right)
    neighbours.append(left)
    neighbours.append(right)
    neighbours.append(down_left)
    neighbours.append(down)
    neighbours.append(down_right)
    return neighbours
```

# B   Artificial intelligence data assimilation

The development of data assimilation has also been revolutionized by artificial intelligence. Xiao et al. [47] were the first to apply the popular traditional numerical data assimilation method, Four-Dimensional Variational, to the AIWP model FengWu. Furthermore, researchers have explored the

development of artificial intelligence assimilation methods, such as FNP [48] and DiffDA [49], which could be applied to both NWP and AIWP models. Although impressive progress has been made, these methods remain limited by reanalysis data and NWP models, which require transforming the observations into physical space. Unlike previous methods, EarthNet [31] proposes implementing observation space data assimilation with masked reconstruction. We are motivated to use an observation AIDA model for formulating a complete observation space.

# C   Comparisons with spatiotemporal learning methods

Table 5: More results on spatiotemporal methods. MAE error of forecasting during 3 lead time periods (0-12h, 12-24h, and 24-36h) for different channels of the satellite data. We use the unit of 1e-5 for AMSU-A, 1e-4 for MHS, and 1e-0 for both ATMS and HIRS.

| Methods | Lead time | AMSU-A | | ATMS | | HIRS | | MHS | |
|---|---|---|---|---|---|---|---|---|---|
| | | ch0 | ch1 | ch0 | ch1 | ch9 | ch10 | ch0 | ch1 |
| Persistence [35] | | 5.86 | 9.15 | 14.37 | 11.69 | 12.43 | 2.21 | 7.01 | 14.74 |
| ConvLSTM [50] | | 127.82 | 208.90 | 73.21 | 78.44 | 77.40 | 11.05 | 62.27 | 158.57 |
| PredRNN [51] | | 2.95 | 4.96 | 6.99 | 6.21 | 9.26 | 1.42 | 4.11 | 10.20 |
| RainFormer [52] | | 3.95 | 6.52 | 9.08 | 8.05 | 10.41 | 1.63 | 4.98 | 11.69 |
| EarthFormer [35] | 0-12h | 18.97 | 33.94 | 37.33 | 38.75 | 22.49 | 3.62 | 18.00 | 55.04 |
| SimVP [53] | | 3.61 | 6.02 | 7.29 | 6.61 | 9.84 | 1.53 | 4.68 | 11.35 |
| TAU [54] | | 3.84 | 6.31 | 7.70 | 6.86 | 9.94 | 1.58 | 4.87 | 11.42 |
| EarthNet [31] | | 2.93 | 4.89 | 6.96 | 6.14 | 9.15 | 1.39 | 3.96 | 9.65 |
| Transformer-DOP [17] | | 2.67 | 4.48 | 6.40 | 5.61 | 9.22 | 1.42 | 3.91 | 9.48 |
| Ours | | 1.92 | 3.39 | 3.36 | 3.27 | 7.70 | 1.12 | 3.07 | 7.91 |
| Persistence [35] | | 4.35 | 6.94 | 10.40 | 8.86 | 13.58 | 2.30 | 6.07 | 15.09 |
| ConvLSTM [50] | | 128.13 | 211.14 | 81.22 | 82.60 | 71.64 | 9.98 | 59.26 | 145.44 |
| PredRNN [51] | | 3.85 | 6.26 | 11.14 | 8.82 | 10.92 | 1.84 | 5.48 | 13.16 |
| RainFormer [52] | | 11.46 | 15.72 | 19.43 | 17.76 | 18.75 | 3.28 | 11.74 | 32.10 |
| EarthFormer [35] | 12-24h | 18.98 | 33.96 | 37.33 | 38.76 | 22.50 | 3.62 | 18.01 | 55.03 |
| SimVP [53] | | 4.83 | 7.74 | 11.48 | 9.10 | 11.49 | 2.02 | 6.43 | 14.20 |
| TAU [54] | | 4.46 | 7.21 | 11.46 | 9.06 | 11.65 | 2.04 | 6.24 | 13.94 |
| EarthNet [31] | | 4.12 | 6.65 | 11.25 | 9.00 | 11.14 | 1.98 | 5.46 | 13.11 |
| Transformer-DOP [17] | | 3.84 | 6.14 | 10.04 | 8.04 | 11.08 | 1.95 | 5.19 | 12.65 |
| Ours  [35] | | 3.11 | 5.12 | 7.35 | 6.35 | 9.57 | 1.54 | 4.51 | 10.54 |
| Persistence [35] | | 6.39 | 9.84 | 15.37 | 12.52 | 14.61 | 2.61 | 8.00 | 17.86 |
| ConvLSTM [50] | | 128.42 | 211.83 | 82.03 | 85.17 | 72.62 | 10.15 | 60.44 | 148.62 |
| PredRNN [51] | | 4.58 | 7.23 | 12.18 | 9.69 | 12.03 | 2.08 | 6.24 | 15.04 |
| RainFormer [52] | | 24.89 | 32.34 | 35.49 | 36.54 | 30.78 | 4.92 | 21.51 | 69.68 |
| EarthFormer [35] | 24-36h | 18.98 | 33.96 | 37.34 | 38.77 | 22.51 | 3.63 | 18.02 | 55.04 |
| SimVP [53] | | 5.72 | 8.91 | 12.93 | 10.33 | 12.71 | 2.32 | 73.95 | 15.99 |
| TAU [54] | | 5.61 | 8.76 | 13.23 | 10.53 | 12.98 | 2.34 | 7.15 | 15.85 |
| EarthNet [31] | | 5.17 | 8.14 | 12.52 | 10.08 | 12.37 | 2.36 | 6.41 | 15.08 |
| Transformer-DOP [17] | | 4.91 | 7.54 | 11.35 | 9.07 | 12.39 | 2.27 | 6.22 | 14.70 |
| Ours | | 3.66 | 5.80 | 7.84 | 6.81 | 10.71 | 1.79 | 5.15 | 12.22 |

As shown in Table 5, we compare DAWP with more spatiotemporal methods including RNN-based ( [50],  [51]), CNN-based ( [53],  [54]), and transformer-based ( [52],  [35]). Our DAWP maintains a significant advantage over these methods, demonstrating the effectiveness of our AIDA module in improving the roll-out and efficiency of AIWP.

We present implementation details of EarthNet and Transformer-DOP here. There is no open-sourced code for EarthNet [31] or Transformer-DOP [17]. For EarthNet [31], it follows the implementation of MultiMAE [32] as detailed in EarthNet's Appendix C and D. Therefore, we reproduce it on our datasets following MultiMAE. As for Transformer-DOP, since the original paper presents only a sketch without details, we implemented it according to our best available understanding. Specifically, EarthNet is reproduced as a 12-layer encoder (hidden dimension 768) paired with an 8-layer decoder (hidden dimension 512), and Transformer-DOP is implemented as a transformer consisting of 18 layers (hidden dimension 1024). We employ sub-images because the full 12-hour global observation sequence would result in 124k-token sequence, which is computationally infeasible.

Table 6: Reconstruction error of various VAEs on different modalities. The column of **Improvement** represents relative average improvement over SD-VAE.

| VAEs | Modality reconstruction error | | | | Improvement |
|---|---|---|---|---|---|
| | AMSU-A (1e-3) | ATMS (1e-2) | HIRS (1e-3) | MHS (1e-2) | |
| SD-VAE [55] | 1.07 | 1.26 | 5.84 | 2.28 | - |
| Mask-SD-VAE | 1.21(-13.1%) | 1.31(-4.0%) | 5.60(+4.1%) | 2.35(-3.1%) | -4.1% |
| ViT-VAE [56] | 0.92(+14.0%) | 1.36(-7.9%) | 4.28(+26.7%) | 2.45(-7.4%) | +6.3% |
| Mask-ViT-VAE | 0.78(+27.1%) | 1.29(-2.3%) | 4.11(+29.6%) | 2.41(-5.7%) | +12.2% |

## D  VAE comparison

We explore the ability of our mask ViT-VAE to compress satellite data with multiple channels and missing values by comparing it with other VAEs.

The results are shown in Table 6, where we compare the reconstruction loss of different VAEs. First, VAEs with ViT structure are more effective for reconstructing modality data with multiple channels, such as HIRS and AMSU-A, while on modalities with fewer channels, there is only a slight increase in reconstruction loss. Another observation is that the application of a mask consistently increases VAEs' reconstruction ability on HIRS. For ViT-VAE, it is beneficial to use the mask for the computation of attention between patch tokens, as it could directly weaken the influence of missing tokens.

## E  Satellite data preprocessing

**Preprocessing**: The original satellite observation data points are extremely sparse and irregular. To spatially align different observation sources and channels for model training, a remapping procedure is performed beforehand. The pseudocode for the preprocessing algorithm is given in algorithm 1.

---

**Algorithm 1:** Remapping Satellite Observation

**Input:** target resolution $R$, observation $D_o$, corresponding latitudes $C_{lat}$ and longitudes $C_{lon}$
**Output:** remapped observation data points $D_{grid}$ on desired global grid

1 Generate global grid $C_R$ of desired resolution $R$ that follows Equirectangular projection;
2 Assign latitudes and longitudes ($C_{lat}$,$C_{lon}$) to the nearest coordinates ($C'_{lat}$,$C'_{lon}$) on grid $C_R$;
3 $D_{grid} \leftarrow$ NaN with the shape of $C_R$;
4 $D_{count} \leftarrow$ Zeros with the shape of $C_R$;
5 **for** $d_o,c'_{lat},c'_{lon}$ *in* $D_o,C'_{lat},C'_{lon}$ **do**
6 $\quad$ Locate corresponding data point $d_{grid}$ of $D_{grid}$ according to coordinates ($c'_{lat},c'_{lon}$);
7 $\quad$ Locate corresponding point counter $d_{count}$ of $D_{count}$ according to coordinates ($c'_{lat},c'_{lon}$);
8 $\quad$ **if** $d_{grid}$ *equals NaN* **then**
9 $\quad\quad$ $d_{grid} \leftarrow d_o$;
10 $\quad$ **else**
11 $\quad\quad$ $d_{grid} \leftarrow (d_{grid} + d_o)$;
12 $\quad$ **end**
13 **end**
14 Average each $d_{grid}$ where $d_{count} \geq 1$

---

**Normalization**: We normalize each modality for efficient convergence. The direct observation modalities are normalized by:

$$x^M_{norm} = \frac{x^M - Mean(x^M)}{Std(x^M)}, \tag{1}$$

where $M$ represents the modality $M$. For ATMS-precipitation, we first implement the log-transformation as:

$$x_{prec} = \log (x/a + b), \tag{2}$$

to alleviate long-tail distribution, and then normalize these variables like other modalities. Specifically, we select $a = 1e - 7$, $b = 1e2$ for SP channel, and choose $a = 1$, $b = 1$ for TCWV channel. The

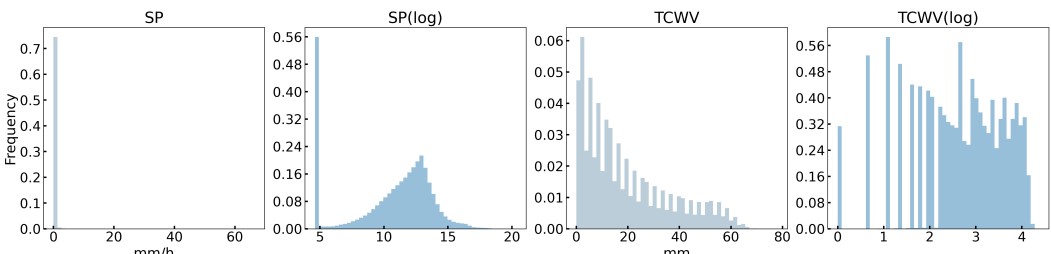

Figure 9: Distribution of ATMS precipitation productions. SP(log) indicates applying a log-transform on SP. It is the same for TCWV(log).

distribution shift is shown in Figure 9. The original distributions also motivate us to choose a threshold list of [0.5 mm/h, 1 mm/h, 2 mm/h] for SP.

## F   Dataset introduction

The four sensors below are organized into a satellite observation dataset for direct observation forecasting. This composite dataset has a training split with data from January of 2012 to June of 2022 for training and a testing split composed of data from May of 2023 to July of 2023. We present detailed introductions to these sensors.

**AMSU-A**: The Advanced Microwave Sounding Unit-A (AMSU-A) is a 15-channel microwave radiometer used for measuring global atmospheric temperature profiles and will provide information on atmospheric water in all of its phases (with the exception of small ice particles, which are transparent at microwave frequencies). AMSU-A will provide information even in cloudy conditions. AMSU-A measures Earth radiance at frequencies (in GHz) as listed under the instrument channel information. Level 1B data was collected from EUMETSAT at `https://archive.eumetsat.int/usc/UserServicesClient.html`.

**ATMS**: The Advanced Technology Microwave Sounder (ATMS) and the Cross-track Infrared Sounder (CrIS) work together to provide global high-resolution profiles of temperature and moisture. These advanced atmospheric sensors create cross-sections of storms and other weather conditions, helping with both short-term nowcasting and long-term forecasting. Level 1C data was collected from GES DISC at `https://disc.gsfc.nasa.gov/datasets?page=1`.

**HIRS**: The High Resolution Infrared Sounder (HIRS) operates at 20 channels (19 channels in the infrared and one in the visible). Its main purpose is to provide input for the vertical temperature and humidity profile retrievals. In addition, the HIRS pixel resolution serves as the standard grid resolution for all ATOVS level 2 products. Level 1B data was collected from EUMETSAT at `https://archive.eumetsat.int/usc/UserServicesClient.html`.

**MHS**: The Microwave Humidity Sounder (MHS) is a 5 channel instrument used to provide input to the retrieval of surface temperatures, emissivities, and atmospheric humidity. In combination with AMSU-A information it can also be used to process precipitation rates and related cloud properties, as well as to detect sea ice and snow coverage. Level 1B data was collected from EUMETSAT at `https://archive.eumetsat.int/usc/UserServicesClient.html`.

**ATMS-Precipitation**: The ATMS-Precipitation is one of the products of the Global Precipitation Measurement (GPM) mission. It is based on the L1C-level calibrated brightness temperature data of the ATMS sensor and extracts information such as precipitation rate and precipitation type through a physical inversion algorithm. Level 2A data was collected from GES DISC at `https://disc.gsfc.nasa.gov/datasets?page=1`.

## G   Training details

Our DAWP framework is trained in 3 stages on 4 A100 80G GPUs, including training mask ViT-VAEs for encoding and mapping, an MMAE for data assimilation in observation space, and a spatiotemporal transformer for direct observation prediction. Specifically, these modules are all trained within 144×144 sub-images. The encoder and decoder of the mask ViT-VAE use the same

Table 7: Hyperparameters for training the mask ViT-VAE of DAWP on the composite dataset.

| Hyper-parameter | Value |
|---|---|
| Learning rate | 0.0001 |
| $\beta_1$ | 0.9 |
| $\beta_2$ | 0.999 |
| Weight decay | 0.00001 |
| Batch size | 200 |
| Training steps | 200000 |
| Warm up percentage | 10% |
| Warmup learning rate | 0.000001 |
| Learning rate decay | Cosine |
| Min learning rate | 0.000001 |
| KL-loss weight | 0.000001 |

Table 8: Hyperparameters for training the AIDA module of DAWP on the composite dataset.

| Hyper-parameter | Value |
|---|---|
| Learning rate | 0.0001 |
| $\beta_1$ | 0.9 |
| $\beta_2$ | 0.999 |
| Weight decay | 0.00001 |
| Batch size | 48 |
| Training steps | 200000 |
| Warm up percentage | 10% |
| Warmup learning rate | 0.000001 |
| Learning rate decay | Cosine |
| Min learning rate | 0.000001 |

Table 9: Hyperparameter for training the AIWP module of DAWP on the composite dataset.

| Hyper-parameter | Value |
|---|---|
| Learning rate | 0.0001 |
| $\beta_1$ | 0.9 |
| $\beta_2$ | 0.999 |
| Weight decay | 0.00001 |
| Batch size | 8 |
| Training steps | 200000 |
| Warm up percentage | 10% |
| Warmup learning rate | 0.000001 |
| Learning rate decay | Cosine |
| Min learning rate | 0.000001 |

transformer structure with a patch size of 16 and a hidden dimension of 768. It is trained with a reconstruction MAE loss and a KL loss weight of 0.000001 for robust representation. We freeze the pretrained mask ViT-VAE as the encoders for each modality in our MMAE. Each modality observation with a 12h time window in a 144×144 sub-image is tokenized into 972 spatiotemporal tokens. Thus, our MMAE totally received 3888 tokens. We randomly select 128 observed tokens of them ( 3.3%) to reconstruct the remaining observed tokens via MAE training. Given the 144× 144 sub-images assimilated by MMAE, our spatiotemporal transformer is trained. It is structured with 12 TS spatiotemporal decoupling blocks, whose hidden dimension is 768. The hyperparameters for optimizing these modules are similar. All of them use the AdamW optimizer with $\beta_0 = 0.9$, $\beta_1 = 0.999$, and a learning rate of 0.0001. The learning rate is scheduled by a cosine scheduler, warming up 10k steps, step by step.

In the table 13, we present the computation cost.

Table 10: The details of the mask ViT-VAE model on different satellite datasets. `Conv16 × 16` is the 2D convolutional layer with $16 \times 16$ kernel. The `FFN` consists of two `Linear` layers separated by a `GeLU` activation layer [57]. The operator SamplePosterior samples a latent representation from $\mu$ and $\sigma$ as SD did [55].

| Module | Layer | Resolution | Channels |
|---|---|---|---|
| Input | - | $144 \times 144$ | $c$ |
| PatchEmbed | Conv16 × 16 | $9 \times 9$ | $c \to 768$ |
| | Flatten | $9 \times 9 \to 81$ | 768 |
| | PosEmbed | 81 | 768 |
| Trnasformer Block × 10 | LayerNorm | 81 | 768 |
| | MaskAttention | 81 | 768 |
| | LayerNorm | 81 | 768 |
| | FFN | 81 | 768 |
| Qauntify | TransformerBlock | 81 | 768 |
| | TransformerBlock | 81 | 768 |
| | Concat | 81 | $768 \to 1536$ |
| | Linear | 81 | $1536 \to 8c$ |
| | SamplePosterior | 81 | $8c \to 4c$ |
| | Linear | 81 | $4c \to 768$ |
| Trnasformer Block × 12 | LayerNorm | 81 | 768 |
| | MaskAttention | 81 | 768 |
| | LayerNorm | 81 | 768 |
| | FFN | 81 | 768 |
| Out | Rearrange | $81 \to 9 \times 9$ | 768 |
| | Conv1 × 1 | $9 \times 9$ | $768 \to 256c$ |
| | Rearrange | $9 \times 9 \to 144 \times 144$ | $256c \to c$ |
| | Conv3 × 3 | $144 \times 144$ | $c$ |

# H   Metrics defination

## H.1   CSI and FAR

For the evaluation of global precipitation variables, the metrics include the Critical Success Index (CSI) and False Alarm Ratio (FAR). They are core binary classification evaluation metrics that quantify the detection accuracy and reliability of precipitation events. In the field of meteorology, these metrics assess the consistency and accuracy between precipitation predictions and observed results, quantitatively evaluating the performance of models. To measure the accuracy of prediction for precipitation with different intensities. Before calculating these metrics, we transform the predicted pixel values and ground truth into binary values (0 or 1) using a given threshold $\tau$. The value is set to 0 if it is less than $\tau$; otherwise, it is set to 1. These binary values enable us to determine the true positive (TP), false positive (FP), false negative (FN), and true negative (TN) counts. CSI, HSS, and FSS are calculated by these counts as follows:

1) Critical Success Index. CSI is a metric that evaluates the proportion of correctly predicted events of hits among conditions, including hits (TP), false alarms (FN), and misses (FP). The formulation of CSI is:

$$CSI = \frac{TP}{TP + FN + FP} \tag{3}$$

The value of CSI ranges from 0 to 1. Higher values indicate better prediction accuracy.

2) False Alarm Ratio. The FAR metric quantifies the proportion of predicted positive events that were actually negative in meteorological verification, emphasizing the reliability of alarm triggers. It is

Table 11: The details of the MMAE on encoded multimodal tokens within a sub-image in a time window 12. `Conv1 × 1` is the 2D convolutional layer with $1 \times 1$ kernel. $(c_1, c_2, c_3, c_4)$ means a multimodal input list with input channels $c_1$, $c_2$, $c_3$, and $c_4$. The MaskTokens is similar to the function of random_mask in [34], while adding the [EOS] tokens to keep the sequences from different samples the same length. The operator of PaddingTokens fills the feature map as [34] did. The FFN consists of two `Linear` layers separated by a `GeLU` activation layer [57].

| Module | Layer | Resolution | Channels |
|---|---|---|---|
| Multimodal Input | - | $9 \times 9 \times 12$ | $(c_1, c_2, c_3, c_4)$ |
| Multimodal PatchEmbed | Conv1 × 1 | $9 \times 9 \times 12$ | $(c_1, c_2, c_3, c_4) \rightarrow (768, 768, 768, 768)$ |
| | Flatten | $9 \times 9 \times 12 \rightarrow 81 \times 12$ | $(768, 768, 768, 768)$ |
| | PosEmbed | $81 \times 12$ | $(768, 768, 768, 768)$ |
| | TemporalEmbed | $81 \times 12$ | $(768, 768, 768, 768)$ |
| | Rearrange | $81 \times 12 \rightarrow 3888$ | $(768, 768, 768, 768) \rightarrow 768$ |
| Random Masking | MaskTokens | $3888 \rightarrow 128$ | 768 |
| Trnasformer Block × 12 | LayerNorm | 128 | 768 |
| | MaskAttention | 128 | 768 |
| | LayerNorm | 128 | 768 |
| | FFN | 128 | 768 |
| Feature Map Filling | Linear | 128 | $768 \rightarrow 512$ |
| | PaddingTokens | $128 \rightarrow 3888$ | 512 |
| | Rearrange | $3888 \rightarrow 81 \times 12 \times 4$ | 512 |
| | PosEmbed | $81 \times 12 \times 4$ | 512 |
| | TemporalEmbed | $81 \times 12 \times 4$ | 512 |
| | Rearrange | $81 \times 12 \times 4 \rightarrow 3888$ | 512 |
| Trnasformer Block × 8 | LayerNorm | 3888 | 512 |
| | Attention | 3888 | 512 |
| | LayerNorm | 3888 | 512 |
| | FFN | 3888 | 512 |
| Multimodal Out | Rearrange | $3888 \rightarrow 972$ | $(512, 512, 512, 512)$ |
| | LayerNorm | 972 | $(512, 512, 512, 512)$ |
| | Linear | 972 | $(c_1, c_2, c_3, c_4)$ |
| | Rearrange | $972 \rightarrow 9 \times 9 \times 12$ | $(c_1, c_2, c_3, c_4)$ |

Table 12: The details of our AIWP module training in the assimilated space. The inputs of this module are sub-images with 8 neighbours in a multimodal way. $(c_1, c_2, c_3, c_4)$ means a multimodal input list with input channels $c_1$, $c_2$, $c_3$, and $c_4$. $Tview$ and $Sview$ indicate the temporal dimension and the spatial dimension as the sequence, respectively. The `FFNwithSwiGLU` consists of two `Linear` layers separated by a `SwiGLU` activation layer [58].

| Module | Layer | Resolution | Channels |
|---|---|---|---|
| Input with Conditions | - | $27 \times 27 \times 12$ | $(c_1, c_2, c_3, c_4)$ |
| PatchEmbed | Concat | $27 \times 27 \times 12$ | $(c_1, c_2, c_3, c_4) \rightarrow c_1 + c_2 + c_3 + c_4$ |
| | Linear | $27 \times 27 \times 12$ | $c_1 + c_2 + c_3 + c_4 \rightarrow 768$ |
| | Rearrange | $27 \times 27 \times 12 \rightarrow 729 \times 12$ | 768 |
| | PosEmbed | $729 \times 12$ | 768 |
| TS Block × 12 | Tview | $729 \times 12 \rightarrow (729 \times) 12$ | 768 |
| | Attention | $(729 \times) 12$ | 768 |
| | LayerNorm | $(729 \times) 12$ | 768 |
| | FFNwithSwiGLU | $(729 \times) 12$ | 768 |
| | LayerNorm | $(729 \times) 12$ | 768 |
| | Sview | $(729 \times) 12 \rightarrow (12 \times) 729$ | 768 |
| | Attention | $(12 \times) 729$ | 768 |
| | LayerNorm | $(12 \times) 729$ | 768 |
| | FFNwithSwiGLU | $(12 \times) 729$ | 768 |
| | LayerNorm | $(12 \times) 729$ | 768 |
| Multimodal Out | Rearrange | $(12 \times) 729 \rightarrow 7 \times 27 \times 12$ | 768 |
| | LayerNorm | $7 \times 27 \times 12$ | 768 |
| | Linear | 972 | $(c_1, c_2, c_3, c_4)$ |

Table 13: Computation cost during inference.

|  | Inference time(ms) | Parameters(MB) | Memory(MB) | Batch size(per GPU) |
|---|---|---|---|---|
| Mask-ViT-VAE | 53 | 96 | 7262 | 50 |
| AIDA | 310 | 105 | 18242 | 12 |
| AIWP | 491 | 216 | 47134 | 2 |

defined as:

$$FAR = \frac{FP}{FP + TP} \tag{4}$$

where FP denotes false positive predictions (e.g., forecasted rainfall with no ground observation) and TP represents true positives (correctly predicted rainfall events). FAR ranges from 0 (perfect reliability) to 1 (all alarms are false), with lower values indicating better prediction specificity.

## H.2 MAE

To evaluate the accuracy of direct observation predictions, we use a pointwise Mean Absolute Error (MAE) as the metric to calculate errors on the ground truth with variable missing values. It is worth noting that the MAE is calculated with the raw observation point by point to ignore the influence of missing values. It is defined as:

$$MAE = \frac{1}{N} \sum_{i=1}^{N} |y_i - \hat{y}_i| \tag{5}$$

N is the total number of points with observation, $y_i$ is the ground truth at the $i^{th}$ location, and $\hat{y}_i$ is the prediction.

# I More results of direct observation predictions

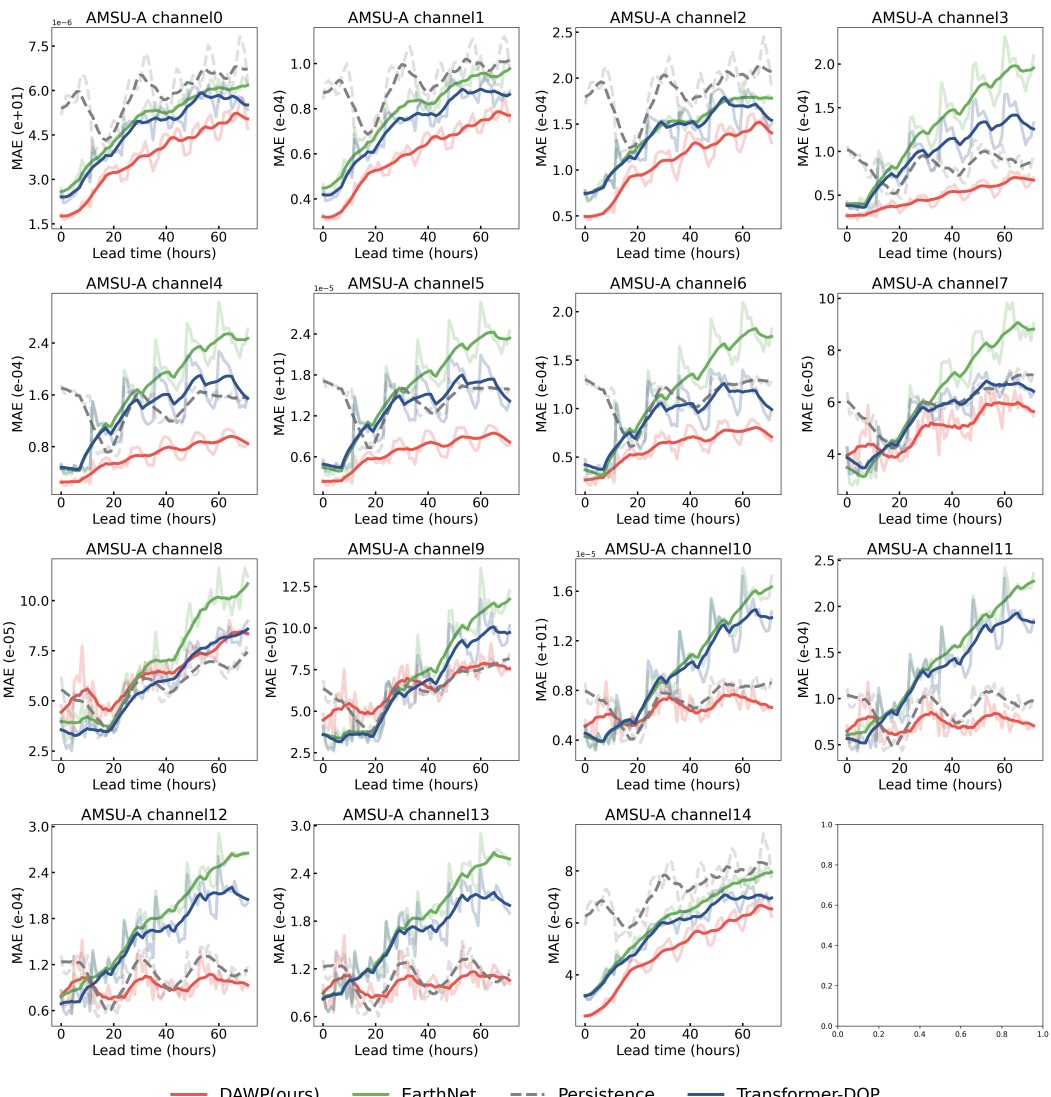

Figure 10: Curves of MAE for the prediction of different channels in sensor AMSU-A. The max leadtime is 72h with a 1h temporal resolution.

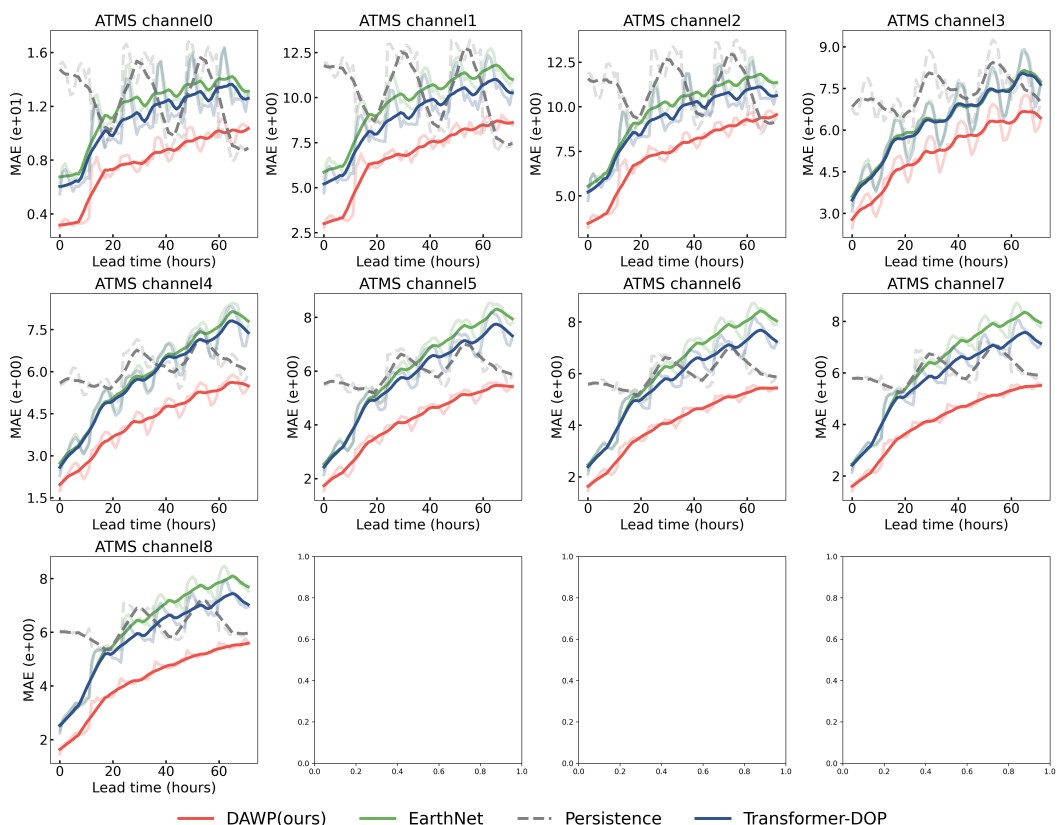

Figure 11: Curves of MAE for the prediction of different channels in sensor ATMS. The max leadtime is 72h with a 1h temporal resolution.

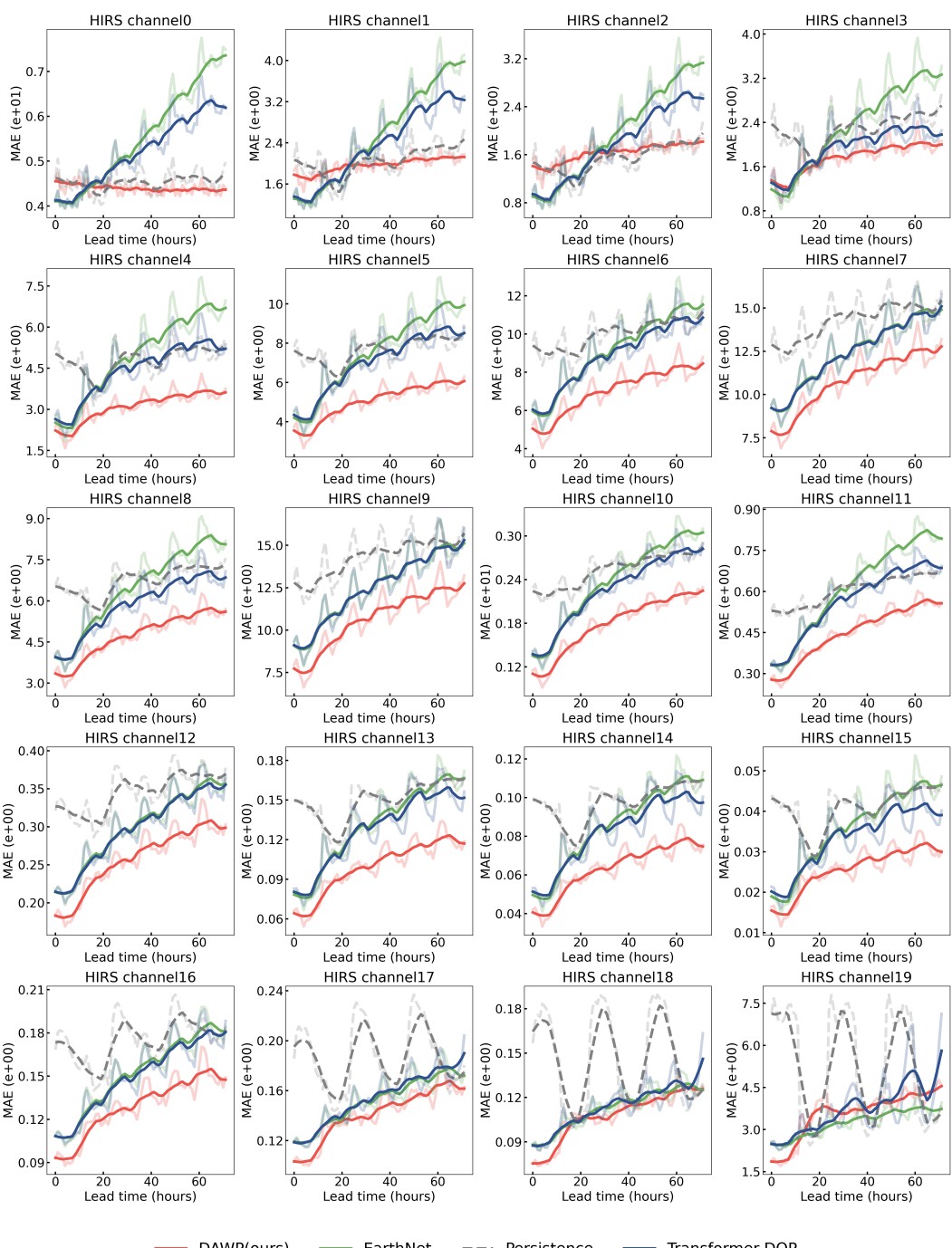

Figure 12: Curves of MAE for the prediction of different channels in sensor HIRS. The max leadtime is 72h with a 1h temporal resolution.

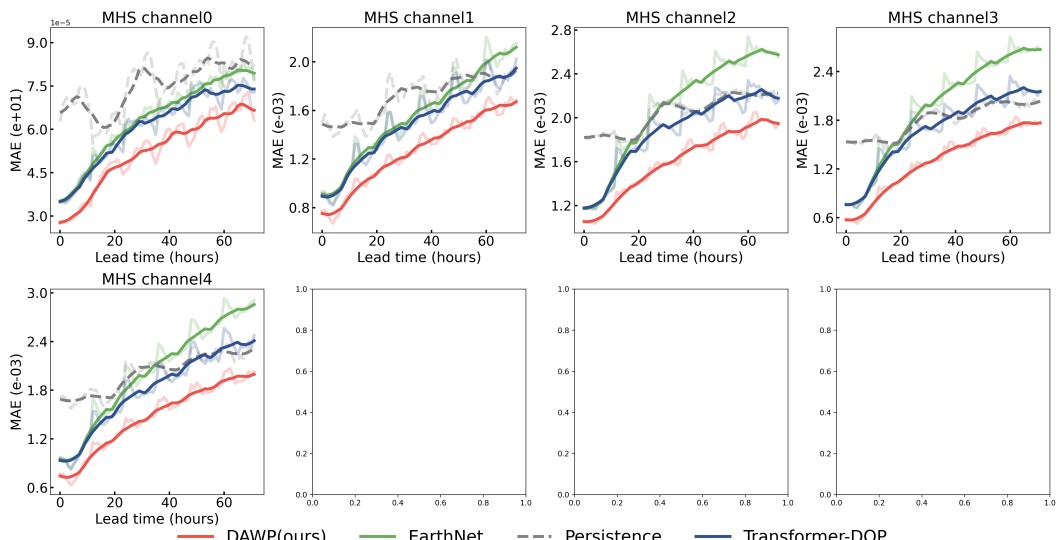

Figure 13: Curves of MAE for the prediction of different channels in sensor MHS. The max leadtime is 72h with a 1h temporal resolution.

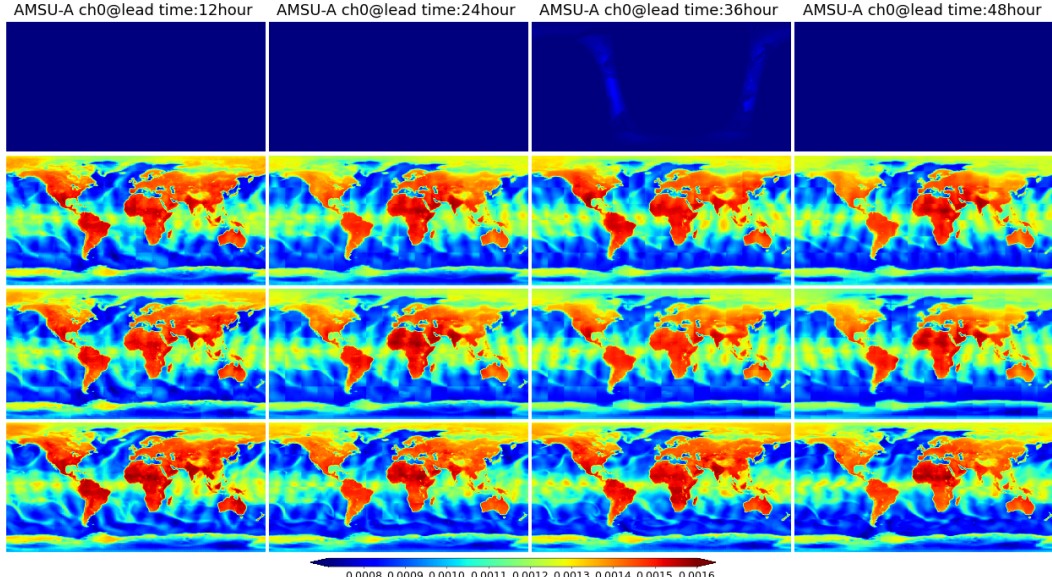

Figure 14: A visualization of rollout predictions for channel 0 of AMSU-A. From top to bottom are the results of ground truth, Transformer-DOP, EarthNet, and DAWP (ours).

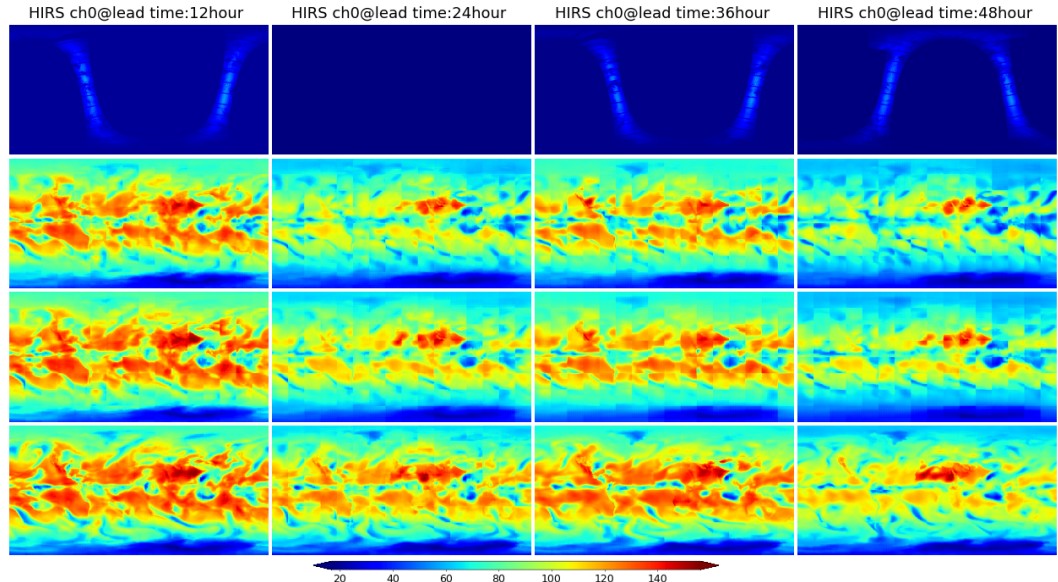

Figure 15: A visualization of rollout predictions for channel 9 of HIRS. From top to bottom are the results of ground truth, Transformer-DOP, EarthNet, and DAWP (ours).

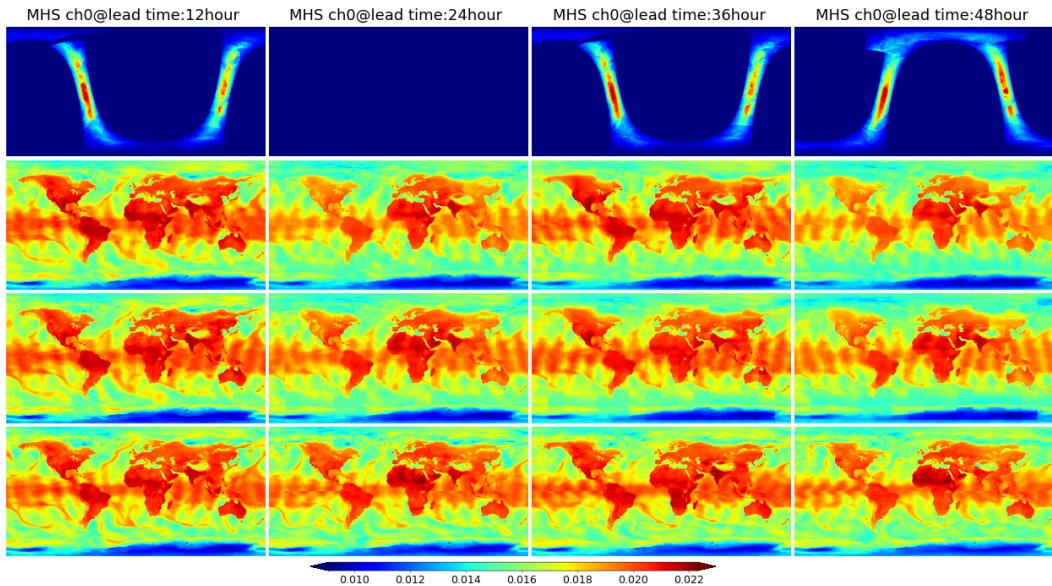

Figure 16: A visualization of rollout predictions for channel 0 of MHS. From top to bottom are the results of ground truth, Transformer-DOP, EarthNet, and DAWP (ours).

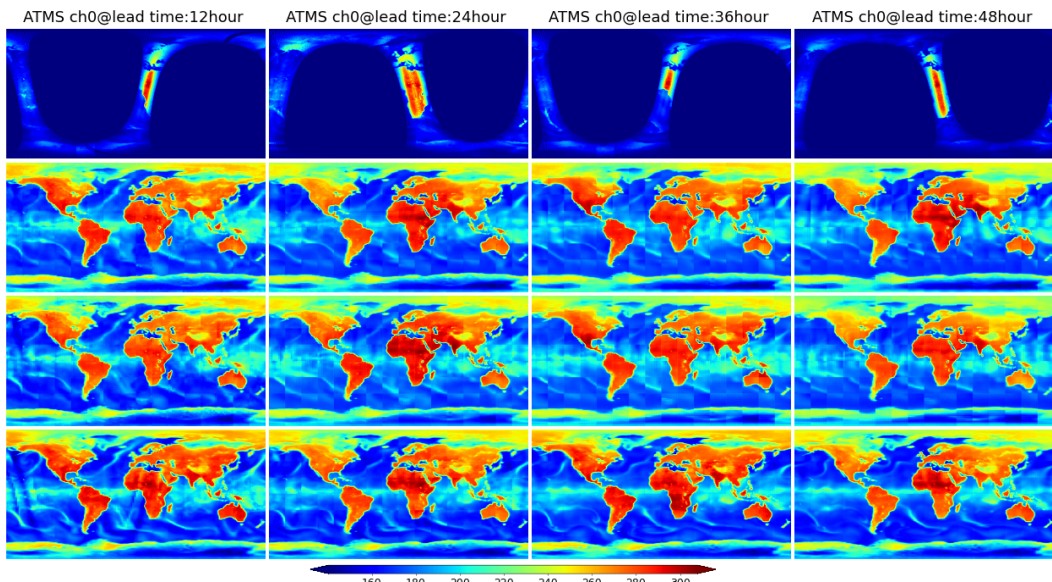

Figure 17: A visualization of rollout predictions for channel 1 of ATMS. From top to bottom are the results of ground truth, Transformer-DOP, EarthNet, and DAWP (ours).

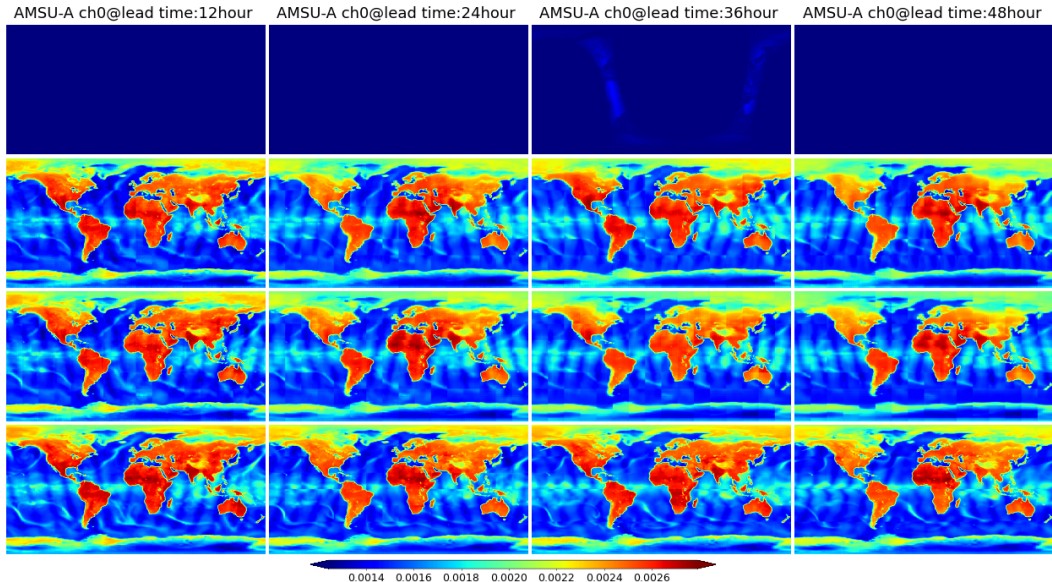

Figure 18: A visualization of rollout predictions for channel 1 of AMSU-A. From top to bottom are the results of ground truth, Transformer-DOP, EarthNet, and DAWP (ours).

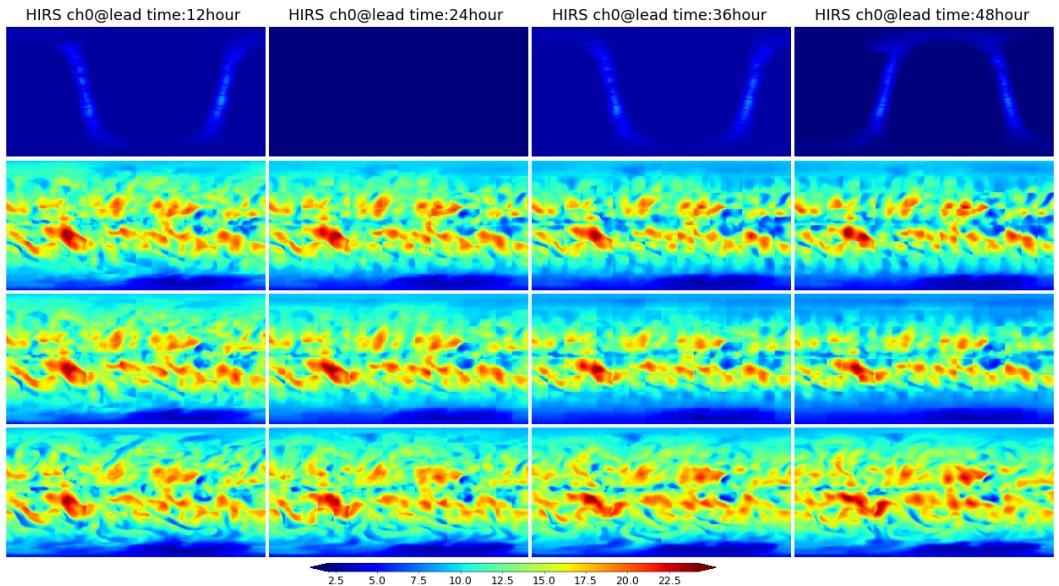

Figure 19: A visualization of rollout predictions for channel 10 of HIRS. From top to bottom are the results of ground truth, Transformer-DOP, EarthNet, and DAWP (ours).

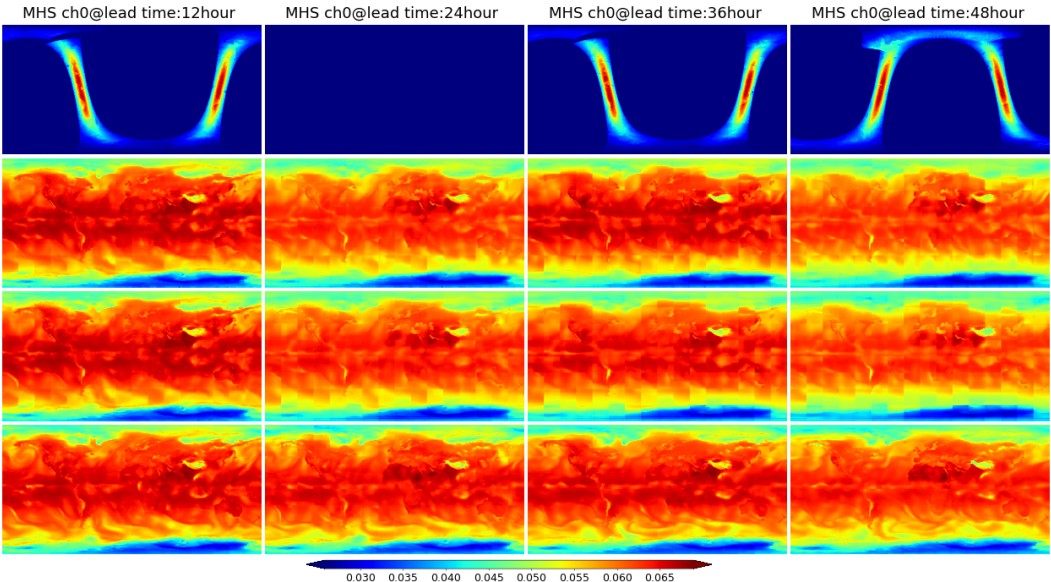

Figure 20: A visualization of rollout predictions for channel 1 of MHS. From top to bottom are the results of ground truth, Transformer-DOP, EarthNet, and DAWP (ours).

