# OpenReview forum: "DAWP: A framework for global observation forecasting via Data Assimilation and Weather Prediction in satellite observation space"
_NeurIPS.cc/2025/Conference — NeurIPS 2025 poster_

### Official Review · Reviewer_bPKH · 2025-06-15

**Clarity:** 2
**Significance:** 4
**Originality:** 3
**Rating:** 5
**Confidence:** 3

**Summary:**

This paper presents DAWP, a framework for global observation forecasting via Data Assimilation and Weather Prediction in satellite observation space. The authors propose to integrate the AIDA module with an AIWP module to enable AIWPs to operate in a complete observation space. Experiments demonstrate that DAWP significantly improves the rollout and efficiency of AIWP and holds promising potential for global precipitation forecasting.

**Questions:**

Have you considered incorporating explicit physical constraints or laws into your data-driven framework to enhance the physical plausibility of the predictions?
How do you address the issue of model explainability in your data-driven approach? Given the complexity of weather systems, how can you ensure that the model’s predictions are not only accurate but also interpretable and trustworthy?

**Ethical Concerns:**

["NO or VERY MINOR ethics concerns only"]

**Final Justification:**

My concerns are well addressed.

**Limitations:**

The author should expand and explain the potential negative impacts of the risk of misuse of the technology or the potential for creating misleading information.

**Quality:**

3

**Strengths And Weaknesses:**

The paper proposes a novel framework that combines data assimilation and weather prediction in a unified observation space, addressing the limitations of existing AIWP methods that rely on reanalysis data. The proposed modules are effective. The paper's contribution to the field of weather prediction is not incremental but rather innovative.
The authors claim that "AIWP models do not require physical solvers to predict the evolution of the atmosphere," which highlights the data-driven nature of their approach. However, this does not preclude the incorporation of physical information into the model. In fact, integrating physical insights could enhance the model's explainability and reliability, potentially preventing it from overfitting to spurious patterns in the data.

---

> ### Author Rebuttal · Authors · 2025-07-31
>
> We sincerely thank you for your insightful and encouraging feedback. We greatly appreciate your recognition of the novelty and significance of our proposed DAWP framework, particularly the integration of the AIDA and AIWP modules to enable forecasting directly in satellite observation space. We are particularly encouraged by your assessment that our contribution is **not merely incremental but genuinely innovative**, as this reinforces the impact of our work.
>
> ----
>
> > **Response to Q1: Incorporation with Physical Constraints**
>
> We sincerely appreciate your valuable suggestion and fully agree that incorporating physical information into the system could enhance performance—particularly in data-limited scenarios—by helping mitigate overfitting.
>
> However, in our current experimental framework, DAWP is trained exclusively on satellite-observed variables—such as microwave radiance, brightness temperature, and infrared radiance—rather than on derived physical quantities like temperature or wind. Because no well‑established physical forward model adequately describes the dynamics of these observed variables, this limitation underscores the need for purely data‑driven approaches in satellite‑observation forecasting.
>
> Moving forward, we see promising opportunities to introduce physical insights gradually. For example, we can incorporate directly measured physical quantities (e.g. wind, atmospheric temperature profiles) into our pipeline and apply Physics-Informed Neural Networks (PINNs) or similar techniques to learn physical relationships, linking these profiles with satellite-observed radiance.
>
> We will make these future directions explicit in the revised manuscript to clarify both the current limitations and the potential for integrating physical priors into DAWP.
>
> > **Response to Q2: Trustworthiness and Explainability**
>
> Thank you for your valuable feedback regarding the credibility and interpretability of data-driven forecasting methods.
>
> While interpretability and trustworthiness remain fundamental challenges for data-driven approaches, DAWP provides an additional perspective on model trustworthiness compared to existing methods such as [1–3], where the completion of sparse observations cannot be supervised. Specifically, during the AIDA initialization stage, DAWP evaluates input completion quality by reconstruction loss over the masked regions, thus enabling a supplementary mechanism for evaluating the credibility of model predictions.
>
> We have also conducted an initial investigation into explainability of forecasting results. In our Modality Sensitivity Analysis, we examine how different input modalities influence the model’s outputs. We observe that using only ATMS data retains most of the model’s predictive power across other modalities, indicating that Brightness Temperature inherently contains substantial information from Microwave and Infrared radiance.
>
> These experiments demonstrate that Brightness Temperature carries rich cross-modal signals, supporting the model’s robustness even when trained with limited modalities.
>
> We believe that interpretability and credibility in direct-observation forecasting remain rich areas for further exploration.
> ____
>
> References:
>
> [1] McNally, Anthony, et al. "Data driven weather forecasts trained and initialised directly from observations." arXiv preprint arXiv:2407.15586 (2024).
>
> [2] Alexe, Mihai, et al. "GraphDOP: Towards skilful data-driven medium-range weather forecasts learnt and initialised directly from observations." arXiv preprint arXiv:2412.15687 (2024).
>
> [3] Vandal, Thomas J., et al. "Global atmospheric data assimilation with multi-modal masked autoencoders." arXiv preprint arXiv:2407.11696 (2024).

---

> > ### Comment · Reviewer_bPKH · 2025-08-01
> > **Concerns are addressed.**
> >
> > The rebuttal makes it clear that the lack of physical priors is not simply an oversight, but a consequence of the data modality (raw radiances) for which no mature physical forward model exists. This is a technology‐gap rather than a methodological limitation.
> > I appreciate the data-driven method of explainability analysis. While these diagnostics are helpful, they remain post-hoc and cannot, by themselves, guarantee that future predictions are physically consistent or free of spurious extrapolation. As the authors acknowledge, this is a field-wide challenge, not unique to DAWP.
> > Overall, the authors’ responses have largely resolved my concerns.

---

> > > ### Author Response · Authors · 2025-08-01
> > > **Thanks for your kindly reply.**
> > >
> > > Thank you for acknowledging both the necessity (**no mature physical forward model exists**) and the challenging nature of this problem (**a field-wide challenge, not unique to DAWP**) of observation-to-forecast modeling. We will further refine our paper based on the insights from this discussion.

---

### Official Review · Reviewer_RPch · 2025-06-23

**Clarity:** 3
**Significance:** 2
**Originality:** 2
**Rating:** 4
**Confidence:** 4

**Summary:**

Recent data-driven weather prediction methods often rely on reanalysis data, which suffer from issues such as data assimilation biases and temporal discrepancies. As an alternative, observation-based forecasting is gaining attention due to its potential to overcome these limitations. This study focuses on learning the spatiotemporal dynamics of observational data. To this end, the paper proposes DAWP, which leverages the AIDA module to enable AI-based weather prediction to learn within a uniform feature space. The AIDA module specifically addresses the issue of missing data commonly encountered in real-world observations by employing a Masked Autoencoder architecture. Additionally, the paper introduces cross-regional boundary conditioning (CBC) to resolve the boundary issues that typically arise in methods based on reanalysis or grid-structured datasets, which are common in the Climate AI community.

**Questions:**

- I agree with the claim that “Temporal Discrepancies: The temporal lag between direct observation acquisition (nearly real-time) and analysis data generation (up to six hours) severely degrades the quick response ability of AIWP models.” However, further investigation into research that recognizes this as a problem and aims to address it is necessary. For example, studies such as "LT3P" [1] have explored this issue.

- The sentence, “As AIWP models do not require physical solvers to predict the evolution of the atmosphere, there is potential for them to directly predict atmosphere states in real-world observation space,” raises some concerns. Are AIWP models truly independent of physical solvers? Isn’t there potential for them to be used in a complementary manner? This could be considered an overclaim. You can refer to "ClimODE" [2].

- In the “Direct Observation Prediction” subsection in Related work, it would be beneficial to include further discussion on studies that specifically address the challenge of “overcoming the dependency” on reanalysis data. While benchmark datasets like WeatherBench1 [3] and WeatherBench2 [4] have long been established within the Climate AI community, they rely on reanalysis data, which introduces latency in data acquisition—making real-world applications difficult. Including references to studies that raise this concern would strengthen the discussion and provide a more comprehensive perspective.

- In Subsection 3.1, the use of an MAE-based model to handle imputation in observation data is a reasonable choice. However, as the authors emphasize the importance of learning spatio-temporal patterns, it might have been more appropriate to adopt a baseline designed for such data—such as VideoMAE [5], which is tailored for video (i.e., spatio-temporal) inputs rather than static images. Why was VideoMAE not considered or used in this context?

- Precipitation nowcasting is typically evaluated using metrics such as CSI (Critical Success Index) and FAR (False Alarm Ratio). However, it would also be beneficial to include the Probability of Detection (POD) metric in the evaluation.


[1] Park, Young-Jae, et al. "Long-Term Typhoon Trajectory Prediction: A Physics-Conditioned Approach Without Reanalysis Data." International Conference on Learning Representations (ICLR), 2024.

[2] Verma, Yogesh, et al "ClimODE: Climate and Weather Forecasting with Physics-informed Neural ODEs." International Conference on Learning Representations (ICLR), 2024.

[3] Rasp, Stephan, et al. "WeatherBench: a benchmark data set for data‐driven weather forecasting." Journal of Advances in Modeling Earth Systems 12.11 (2020): e2020MS002203.

[4] Rasp, Stephan, et al. "Weatherbench 2: A benchmark for the next generation of data‐driven global weather models." Journal of Advances in Modeling Earth Systems 16.6 (2024): e2023MS004019.

[5] Wang, Limin, et al. "Videomae v2: Scaling video masked autoencoders with dual masking." Proceedings of the IEEE/CVF conference on computer vision and pattern recognition (CVPR), 2023.

**Ethical Concerns:**

["NO or VERY MINOR ethics concerns only"]

**Final Justification:**

The rebuttal provides clear and convincing clarifications that fully resolve my earlier concerns. The authors have effectively restated their core contribution, provided rigorous comparisons with relevant prior work, and offered well-justified explanations for key design choices such as the AIDA–AIWP framework and their approach to handling sparse observational inputs. The inclusion of additional references, expanded evaluation metrics (including POD), and thoughtful discussion on both methodological and real-world aspects further enhance the manuscript. The experiments are solid and demonstrate a genuine effort to address recognized challenges within the climate community. Based on these improvements and clarifications, I am increasing my score from 3 to 4.

**Limitations:**

- It is encouraging that this study collects a dataset, constructs it thoughtfully, and establishes baseline models to conduct diverse and robust experiments. However, it would be beneficial if the specific problem the study aims to solve were more clearly discussed in both the Introduction and Conclusion, and explicitly connected to the baselines.

**Paper Formatting Concerns:**

- This study clearly follows the paper formatting guidelines.

**Quality:**

3

**Strengths And Weaknesses:**

## Strengths
- The dataset and experimental setup are robust. In particular, the experiments effectively highlight the individual importance of each component within the dataset and baseline models. Moreover, as shown in Section 1 of the Supplementary Material, the study includes comparisons with existing video prediction models from the computer vision (CV) community, thereby addressing the interests and questions of both the Climate AI and the ML/CV communities.
- The "Ablation Study: Gains of Cross-Regional Boundary Conditioning" section, including Figure 7 and the Supplementary Material, represents the strongest aspect of this study. Most existing data-driven methods for weather forecasting rely on gridded datasets, which often produce unnatural results near boundaries. In contrast, this work effectively mitigates that issue to a significant extent.
- The study effectively leverages multiple modalities or sensors to address the problem. In particular, the use of a Masked Autoencoder (MAE) to tackle the common issue of dataset imputation in real-world scenarios is both timely and appropriate.

## Weaknessess
- The limitations of prior studies and the motivation presented in the Introduction are framed in a way that aligns with general themes commonly discussed in the Climate AI community, but they lack specificity. The mentioned issues, such as data assimilation biases and temporal discrepancies, are also rather generic. Notably, the discussion on temporal discrepancies is not addressed beyond the Abstract and Introduction, which weakens the clarity and depth of the problem statement.

---

> ### Author Rebuttal · Authors · 2025-07-31
>
> We sincerely thank you for your thoughtful and encouraging feedback. We greatly appreciate your recognition of the importance of **observation-based forecasting as a promising direction** beyond reanalysis data, as well as your positive assessment of our AIDA module and cross-regional boundary conditioning (CBC) approach. Your comments on the robustness of our experimental design, especially the detailed ablation studies and cross-community comparisons, are highly motivating. We are particularly grateful for your acknowledgment of our efforts to address real-world challenges such as missing data and boundary artifacts through well-justified architectural choices.
>
> **OUR CORE CONTRIBUTION:**
>
>
> We **identify a key challenge** in observation-based forecasting (L55): the **mismatch between sparse observations and dense prediction**. Our proposed DAWP framework effectively mitigates the input–output mismatch by introducing a **two-stage solution**—AIDA for assimilation-based initialization, followed by an AIWP forecasting model specifically adapted to AIDA outputs. As shown in Figure 6, our results provide strong empirical evidence supporting this claim.
>
> ----
>
>
> > **Response to Q1: Discussion about LT3P**
>
> Thank you for drawing our attention to LT3P as a relevant real‑time reference. We will include LT3P in our “Related Work” section. As you point out, LT3P leverages reanalysis data for pre‑training and combines NWP forecasts with historical typhoon trajectories to make real‑time predictions.
>
> In contrast, we would like to clarify that our DAWP framework intentionally avoids reanalysis data and NWP systems, instead focusing exclusively on observation-driven methods. The primary contribution of DAWP is to demonstrate that the combined AIDA–AIWP framework offers an effective solution to the mismatch between observational input space and the forecasting objective.
>
> We still believe that the approach adopted in LT3P—pretraining on ERA5 reanalysis data followed by fine-tuning on real-time data—also provides valuable inspiration for future extensions of our framework.
>
>
> > **Response to Q2: Relationship of AIWP and Physical Solver**
>
> Thank you for bringing ClimODE to our attention and enabling a more focused discussion.
>
> **Network**: ClimODE leverages the advection equation to design a prediction network trained on reanalysis data. While it does incorporate physical priors via a physics‑informed Neural ODE, its forecasting performance remains well below that of the numerical weather prediction system IFS (see Figure 4 of ClimODE). By contrast, methods such as Pangu‑Weather[1], GraphCast[2], and GenCast[3] are also trained on reanalysis datasets but do not incorporate any explicit physical solver. Despite the lack of physical constraints, they consistently outperform IFS across a wide range of performance metrics and lead times. This suggests that including a physics-based solver in network design is not strictly necessary for achieving superior forecast skill.
>
> **Data**: Moreover, the use of reanalysis data itself implicitly introduces physical solver dependence, since such data are generated using NWP systems. In fact, purely observation‑driven models also show remarkable performance in some domains—for example, precipitation nowcasting[4]. Thus, from both network design and data selection perspectives, AIWP models operate independently of physical solvers.
>
> Nevertheless, we fully agree with you that complementary integration of physical priors could be beneficial especially in scenes without enough data. Such priors can be incorporated either by training with reanalysis data or by applying Physics‑Informed Neural Networks (PINNs) to observational physical variables. In future versions of our manuscript, we will expand on these points and provide clearer justification, thereby avoiding any potential overclaim.
>
>
> > **Response to Q3: More References**
>
> Thank you very much for your kind suggestion regarding WeatherBench. Since the ERA5 reanalysis dataset serves as the foundation for WeatherBench, it is an excellent reference to raise the concern of reanalysis. Additionally, beyond the references already included namely EarthNet[5], Transformer-DOP[6], and GraphDOP[7]—we will also incorporate references of OMG-HD[8] and LT3P, which further support the concerns raised.
>
>
> > **Response to Q4: VideoMAE**
>
> Thank you for suggesting VideoMAE—this enables a more precise discussion.
>
> VideoMAE’s tube masking strategy creates spatiotemporal tokens by masking the same spatial location across multiple frames to exploit temporal redundancy. However, due to the polar-orbiting satellite motion characteristic, consecutive frames often lack overlapping observations at the same location. If we employed VideoMAE’s tube masking, the resulting tokens would contain empty values, forcing the model to learn from missing data—an undesirable artifact in our context. Consequently, we did not adopt this type of MAE design that relies on temporal redundancy between neighboring frames.
>
> > **Response to Q5: Evaluating POD**
>
> Thank you for your suggestion. We have evaluated the POD metric, and the conclusions are consistent with those presented in Table 3—our proposed DAWP framework significantly outperforms the other methods.
>
> |                 | TCWV (mm)   |             |             | SP (mm/h)   |             |             |
> |-----------------|-------------|-------------|-------------|-------------|-------------|-------------|
> |                 | POD-10      | POD-20      | POD-30      | POD-0.5     | POD-1.0     | POD-2.0     |
> | EarthNet        | 0.948 | 0.923 | 0.915 | 0.171 | 0.053 | 0.009 |
> | Transformer-DOP | 0.948 | 0.924 | 0.913 | 0.175 | 0.062 | 0.011 |
> | Ours            |     **0.951**        |      **0.926**       |      **0.917**       |     **0.257**        |    **0.121**         |      **0.039**       |
>
>
> >**Response to W1: Motivations and In-depth Analysis about Challenges**
>
> We would like to re-clarify **OUR CORE CONTRIBUTION**, as presented at the beginning of the rebuttal.
>
> We agree with you that the limitations of reanalysis data are a consensus within the community. As a result, Direct Observation-based Prediction have recently gained significant attention, given their potential to overcome those limitations.
>
> Our main contribution lies in first identifying the mismatch between sparse inputs and dense predictions as a key challenge within DOPs. To mitigate such mismatch, we propose a novel AIDA–AIWP framework. Our experimental results provide strong support for the effectiveness of this framework in resolving the mismatch issue as shown in figure 5, highlighting its value as a step forward in DOP research.
>
> ___
>
> We sincerely appreciate your feedback, which has significantly improved the paper’s clarity and completeness. Should further clarification be required, please do not hesitate to reach out, and we will address your concerns promptly. We look forward to your continued input.
>
> References:
>
> [1] Bi, Kaifeng, et al. "Accurate medium-range global weather forecasting with 3D neural networks." Nature 619.7970 (2023): 533-538.
>
> [2] Lam, Remi, et al. "Learning skillful medium-range global weather forecasting." Science 382.6677 (2023): 1416-1421.
>
> [3] Price, Ilan, et al. "Probabilistic weather forecasting with machine learning." Nature 637.8044 (2025): 84-90.
>
> [4] Gao, Zhihan, et al. "Earthformer: Exploring space-time transformers for earth system forecasting." Advances in Neural Information Processing Systems 35 (2022): 25390-25403.
>
> [5] Vandal, Thomas J., et al. "Global atmospheric data assimilation with multi-modal masked autoencoders." arXiv preprint arXiv:2407.11696 (2024).
>
> [6] McNally, Anthony, et al. "Data driven weather forecasts trained and initialised directly from observations." arXiv preprint arXiv:2407.15586 (2024).
>
> [7] Alexe, Mihai, et al. "GraphDOP: Towards skilful data-driven medium-range weather forecasts learnt and initialised directly from observations." arXiv preprint arXiv:2412.15687 (2024).
>
> [8] Zhao, Pengcheng, et al. "OMG-HD: A high-resolution AI weather model for end-to-end forecasts from observations." arXiv preprint arXiv:2412.18239 (2024).

---

> > ### Comment · Reviewer_RPch · 2025-08-08
> >
> > Thank you for your thorough and well-structured rebuttal, which has fully addressed my concerns. I appreciate the clear articulation of your core contribution, the detailed comparisons with related work such as LT3P and ClimODE, and the thoughtful explanations regarding design choices. Your additional analyses, expanded references, and inclusion of POD evaluation results further strengthen the manuscript and enhance its clarity. I am satisfied that my earlier questions have been resolved, and I believe the revisions and clarifications meaningfully improve the overall quality and impact of the work.

---

> > > ### Author Response · Authors · 2025-08-08
> > >
> > > Dear Reviewer RPch,
> > >
> > > Thank you for your recognition and positive feedback. We are pleased that our revisions have adequately addressed your previous concerns.
> > >
> > > We will carefully revise the manuscript following your suggestions, including a clear articulation of our core contribution in the introduction, adding discussions of related work such as LT3P and ClimODE in the related work, and completing the precipitation evaluation with POD.
> > >
> > > If there are any further questions, feel free to raise them, and we are always ready for the discussion.
> > >
> > > Best regards,
> > >
> > > The Authors

---

> ### Author Response · Authors · 2025-08-05
> **Looking forward to hearing from you**
>
> Dear Reviewer RPch,
>
> Thank you once again for your thoughtful and constructive feedback.
>
> In response to your comments, we first clarify our main contribution—identifying the mismatch between sparse observations and dense predictions in Direct Observation-based Prediction (DOP) and proposing a novel solution through the AIDA–AIWP framework. This two-stage approach effectively mitigates the mismatch issue and is supported by strong experimental evidence. We also evaluate POD metrics to better assess the precipitation forecasting performance of DOP methods. Additionally, we have included discussions on LT3P and ClimODE to provide a more accurate characterization of DOP methods. Furthermore, we explain why VideoMAE is not suitable for satellite observations, due to the unique challenges posed by polar-orbiting satellite motion. Finally, we express our willingness to explore the integration of physical priors in future work.
>
> We are actively participating in the Author-Reviewer Discussion phase and would be happy to provide further clarification if needed! We are committed to addressing all reviewer concerns and ensuring timely responses. Thank you once again for your contributions to our work.
>
> Best regards,
> The Authors of Submission 1085

---

### Official Review · Reviewer_Ht96 · 2025-07-02

**Clarity:** 2
**Significance:** 2
**Originality:** 2
**Rating:** 3
**Confidence:** 4

**Summary:**

This work proposes a novel framework to do the weather forecast directly from the observation space. The framework consists of two steps: (a) Initializing reanalysis data and data assimilation are substituted by an artificial intelligence assimilation module via multi-modality masking, (b) Then a regional boundary conditioning is used to learn the spatio-temporal correlations between irregular and high-resolution satellite observations, resulting in sub-image global observation forecasting.

**Questions:**

Questions:

- Don’t we also lose valuable information if you don’t use reanalysis? Is there any experiments to support the main claim of the manuscript?
- What happens if you remove the analysis observation or the AIDA and predicts directly the output without griding?
- L138: How were these values defined? The fixed time window of 12 and a sub-image of 144×144?
- L175-176: “The ViT encoder/decoder provides better compression capability, as detailed in Appendix C, compared to SD-VAE for satellite observation”, is there any experiments to support this claim?
- How about Graphdop I think it is an important baseline?
- Table 5 looks like the MAE is lower for SD-VAE for higher values while small fluctuation for smaller values like AMSU-A and HIRS. What is the reason and are these results reported with different random seeds?
- L656-687: I don't think this mimics a real operational setting. Are these satellite measurements available at near-real time?

Minor:

- L93: Do we need 35TB, for a 12×1152×2304 resolution?
- Describing the method is very short.
- The manuscript uses a lot of uncommon abbreviation like AIDA, AIWP, and transformer-dop.
- L197: transformer-dop? The reference what model is used here?
- The qualitative results don’t show anything, it is better to have a difference to ground truth as a comparison rather than plotting the absolute value over the full domain.
- Figure 5 looks weird the model doesn’t learn at all if you remove AIDA.

**Ethical Concerns:**

["NO or VERY MINOR ethics concerns only"]

**Final Justification:**

The paper presents a new deep learning model and a new protocol to learn directly from the observation space.

First, the idea of learning directly from observations has been introduced by Vandal et al. 2024 (EarthNet) and McNally et al. 2024 (Transformer-DOP) and followed by GraphDOP by Alexe et al. 2024. I do not think the novelty is high enough regarding this aspect.

Second, in my opinion, the adaptation of the baselines is not suitable and the evaluation can be misleading. The paper gives a false impression that the baselines are the ones from Vandal et al. 2024 and McNally et al. 2024 . However, they were re-implemented and tested in a different setting than their own original protocols (see weaknesses 4 and 5). There is no information about the baselines other than the number of layers which the authors provided during the discussion. There are no details about the baselines in the supplementary too. What I want to say is that whatever baseline is being tested here, it will probably fail and generate artifacts because of the sub-images approach and the new protocol that the baselines were not designed for in the first place. I believe that the baselines should have been evaluated in a fair setting e.g., following their original protocols to generate results.

Given these remaining issues I tend toward a reject.

**Limitations:**

Would the framework work for very sparse observations data? This should be mentioned in the limitation.

**Paper Formatting Concerns:**

The instruction block for the NeurIPS Paper Checklist should be removed.

**Quality:**

2

**Strengths And Weaknesses:**

Strengths:

- This work addresses a critical problem in the weather forecast especially suited for precipitation forecast.
- The idea of the proposed framework is clear for the reader.

Weaknesses:
- The manuscript argues that using reanalysis and data assimilation as initial condition has shortcomings and the main idea of this work is to overcome this issue. However, there was no comparison to baselines trained directly on reanalysis data.
- The paper does not have a validation set but rather optimized directly on the test set. I assume that ablation studies are done on the test set as well instead on the validation one. In addition, the test set is very short and covers only 3 months. In my opinion this split is inadequate and the test set does not ensure a representative evaluation.
- The improvement is not statistically significant (i.e., the results are generated by one random seed).
- The baselines were trained with sub-image and not with their own original implementations. The artifacts on the grid are obvious from Fig. 4.
- Experimental result reproducibility: The manuscript does not provide details about the baselines.
- The manuscript does not provide sufficient information on the computer resources (type of compute workers, memory, time of inference) needed to reproduce the experiments, parameters and model size.

---

> ### Author Rebuttal · Authors · 2025-07-31
>
> We sincerely appreciate your thoughtful and constructive feedback. Thank you for recognizing the novelty and clarity of our proposed framework.
>
> **OUR CORE CONTRIBUTION:**
>
> We **identify a key challenge** in observation-based forecasting: the **mismatch between sparse observations and dense prediction**. DAWP mitigates this mismatch via **a two-stage solution**: AIDA for initialization and AIWP for forecasting. Figure 6 provides empirical evidence.
>
> ---
>
> > **Response to Q1: Main claim of DAWP and Relationship between Reanalysis or Observation**
>
> Thank you for the thoughtful question. Our main claim is that DOPs trained in the space initialized by AIDA improve prediction, as stated in Figure 1, lines L54–56 and **OUR CORE CONTRIBUTION**. Table 2 and Figure 6 support this claim.
>
> We agree that Reanalysis models benefit from human priors in NWP (physical laws, bias correction, assimilation). However, exploring data‑driven forecasting remains scientifically valuable and timely. DOP extends the traditional reanalysis-based forecasting paradigm into a heterogeneous, spatiotemporal observational space, encompassing multiple data modalities—an area that remains underexplored to date.
>
> > **Response to Q2: Results without AIDA.**
>
> Thank you for raising this important point. Results without AIDA are shown in Figures 5 and 6.
>
> **Learning dynamics**: When AIDA is not applied, learning dynamics are unstable for our spatial-temporal prediction model, producing severe spikes in learning phase (Fig.5) due to the irregular and sparse nature of input. AIDA stabilizes learning by mapping sparse inputs into a dense, structured latent space.
>
> **Roll-out**: Without AIDA, roll-out degrades because sparse training inputs mismatch dense inference inputs in roll-out (Fig. 6).
>
> > **Response to Q3: Window Length and Sub-image Size**
>
> A 144×144 sub-image with patch size 16 over 12 steps and 4 domains yields 3884 tokens. This length fits our transformer and computational budget, ensuring efficient attention.
>
> The 12-step window matches the satellite revisit cycle, ensuring global spatial coverage.
>
> > **Response to Q4: Experiments to Support ViT-VAE's Capability**
>
> Thank you for the suggestion. The comparisons are presented in Appendix C. When compared against SD‑VAE, our MaskViTVAE model achieves approximately 12 % relative improvement in reconstruction loss.
>
> > **Response to Q5: GraphDOP**
>
> GraphDOP is an excellent project. With ECMWF's data infrastructure and preprocessing capabilities, it was trained on 32 observation types, broader than  DAWP’s 4 types. Additionally, pecific implementation details are not clearly disclosed. Without access to the codes, the direct comparison is currently infeasible. We are willing to evaluate it using data available to us when GraphDOP is open-sourced.
>
> > **Response to Q6: Reasons for Different Improvments and VAEs with Random Seeds**
>
> In Table 5, performance variation is due to channel count differences, not data scale. All datasets are normalized, so value ranges do not affect results. Gains follow channel counts (HIRS 20 > AMSU-A 15 > ATMS 9 > MHS 5). Our method excels on high-channel data, while SD-VAE is more suited to low-channel cases. Thus, DAWP provides stronger compression for high-dimensional satellite observations.
>
> We also tested four seeds (0–3), retraining models and reporting mean ± std. The results show our method’s gains are robust to random seed choice.
>
>
> ||AMSU-A(1e-3)|ATMS(1e-2)|HIRS(1e-3)|MHS(1e-2)|
> |-|-|-|-|-|
> |SD-VAE|1.08±0.00|1.37±0.02|5.94±0.08|2.40±0.02|
> |Ours|0.76±0.01|1.16±0.05|3.88±0.11|2.48±0.05|
>
>
> > **Response to Q7:  Near-realtime Availability**
>
> We use satellite Level‑1 products, available immediately after transmission; for instance, EUMETSAT[1] provides MHS with <60 min latency.
>
>
> > **Response to W1: Main Idea and Baselines on Reanalysis**
>
> We thank the reviewers and clarify our main points. The core contribution of DAWP is the design of a satellite‑observation prediction framework—comprising AIDA and AIWP—to directly predict satellite observational quantities, rather than comparing with reanalysis-based forecasts.
>
> **Data**: DAWP predicts satellite radiances (microwave, brightness temperature, infrared). In contrast, the reanalysis‑based baselines forecast physical variables like wind and temperature. As these predictions involve different modalities, direct comparison is infeasible.
>
> **Architecture**: Reanalysis-based models lack designs to process sparse, irregular satellite inputs, hence not directly applicable for predicting satellite observables.
>
>
> > **Response to W2&W3: Dataset Split and Results with Random Seeds**
>
> Thank you for your suggestions. Originally: Training: 2012.1 – 2022.6, Validation: 2022.7 – 2023.4, Testing: 2023.5 – 2023.7. All experiments strictly followed this training–validation–testing partitioning protocol. This 3-month test follows GraphDOP (which used only 1 month in Section 5.2) and EarthNet (which used 2 months in Section E.2). Following your suggestion, we revised to: Train 2012.1–2021.7, Val 2021.8–2022.7, Test 2022.8–2023.7.
>
> We retrained all models on the revised split with seeds {0,1,2,3}; results (mean±std) confirm statistical significance. We thank you again and will include this extended evaluation.
>
> |Lead time: 0-12h |ATMS0|ATMS1|AMSU-A0(e-5)|AMSU-A1(e-5)|MHS0(e-4)|MHS1(e-4)|HIRS9|HIRS10|
> |-|-|-|-|-|-|-|-|-|
> |EarthNet|7.09±0.18|6.13±0.15|2.52±0.04|4.35±0.07|3.91±0.04|9.86±0.06|8.60±0.02|1.31±0.01|
> |Transformer-DOP|6.19±0.19|5.46±0.12|2.37±0.01|4.10±0.02|3.78±0.17|9.56±0.01|8.55±0.02|1.31±0.00|
> |Ours|**3.25±0.01**|**3.17±0.00**|**1.76±0.00**|**3.22±0.02**|**3.07±0.02**|**8.06±0.04**|**7.29±0.02**|**1.05±0.00**|
>
> |Lead time: 12-24h|ATMS0|ATMS1|AMSU-A0(e-5)|AMSU-A1(e-5)|MHS0(e-4)|MHS1(e-4)|HIRS9|HIRS10|
> |-|-|-|-|-|-|-|-|-|
> |EarthNet|11.29±0.43|9.06±0.09|4.07±0.09|6.56±0.08|5.35±0.02|13.42±0.06|10.31±0.05|1.85±0.03|
> |Transformer-DOP|10.02±0.37|7.99±0.19|3.65±0.02|5.86±0.05|5.23±0.06|12.94±0.08|10.20±0.05|1.78±0.01|
> |Ours|**7.97±0.34**|**6.59±0.19**|**3.08±0.07**|**5.04±0.09**|**4.48±0.08**|**10.94±0.06**|**9.07±0.04**|**1.48±0.01**|
>
> |Lead time: 24-36h|ATMS0|ATMS1|AMSU-A0(e-5)|AMSU-A1(e-5)|MHS0(e-4)|MHS1(e-4)|HIRS9|HIRS10|
> |-|-|-|-|-|-|-|-|-|
> |EarthNet|12.55±0.49|10.18±0.18|5.13±0.13|8.04±0.11|6.42±0.08|16.05±0.16|11.65±0.11|2.23±0.06|
> |Transformer-DOP|11.48±0.43|8.98±0.25|4.57±0.05|7.06±0.07|6.32±0.07|15.31±0.12|11.36±0.08|2.08±0.09|
> |Ours|**8.73±0.11**|**7.16±0.06**|**3.64±0.07**|**5.73±0.10**|**5.20±0.08**|**13.02±0.13**|**10.26±0.03**|**1.75±0.01**|
>
>
> > **Response to W4&W5: Baselines Implementations**
>
> Thank you for your constructive feedback. There is no open‑sourced code for EarthNet[2] or Transformer‑DOP[3]. For EarthNet, it follows the implementation of MultiMAE[4] as detailed in EarthNet’s Appendix C and D. Therefore, we reproduce it on our datasets following MultiMAE. As for Transformer‑DOP, since the original paper presents only a sketch without details, we implemented it according to our best available understanding. Specifically, EarthNet is reproduced as a 12‑layer encoder (hidden dimension 768) paired with an 8‑layer decoder (hidden dimension 512), and Transformer‑DOP is implemented as a transformer consisting of 18 layers (hidden dimension 1024). We employs sub‑images because the full 12‑hour global observation sequence would result in 124k-token sequence, which is computationally infeasible.
>
> Artifacts (Fig. 4) arise from (1) mismatch between dense autoregressive inference and sparse training and (2) lack of cross‑region info in sub‑images. Our AIDA initialization and CBC module resolve these problems.
>
> > **Response to W6: Computer resources**
>
> Thank you for your suggestion. In our experiments, we used 4 A100 80G GPUs for parallel training. In the table below, we present other details, which would be added in the future version.
> ||Inferennce time(ms)|Parameters(MB)|Memory(MB)|Batch size(per GPU)|
> |-|-|-|-|-|
> |Mask-ViT-VAE|53|96|7262|50|
> |AIDA|310|105|18242|12|
> |AIWP|491|216|47134|2|
>
> > **Response to M1: 35TB Observations**
>
> Each observation is a 12×1152×2304 grid with spectral channels (ATMS‑9, AMSU‑A‑15, MHS‑5, HIRS‑20). With two observations per day over 11 years, the dataset totals ~35 TB.
>
> > **Response to M2: Method Details**
>
> In the Methods section, we emphasize the conceptual pipeline combining AIDA and AIWP to address prediction based on irregular and sparse satellite observations. Model architecture details (layers, dimensions, attention, tokenization) are in Tables 9–11. We will expand methodological descriptions and design rationale in future work.
>
> > **Response to M3: Uncommon Abbreviation**
>
> We would like to clarify that Artificial Intelligence Data Assimilation (AIDA) and Artificial Intelligence Weather Prediction (AIWP) are both expanded and carefully defined in L2 and L11, respectively. Transformer‑DOP denotes a Transformer backbone for the DOP task (Ref. 16). We are open to clearer alternative naming suggestions
> > **Response to M4: Explanations about Transformer–DOP**
>
> Transformer‑DOP refers to Ref. 16. As the paper provides only a high-level description, we reproduced it based on its conceptual design.
> > **Response to M5: Visualization**
>
> Thank you for the suggestion. Our aim is to emphasize the complementation capability of DAWP for handling irregular and sparse satellite observations, especially in comparison to other methods that suffer from artifacts arising from sub-image tiling and mismatched input-output density. In future versions, we will incorporate visualizations of prediction–truth differences.
>
> ___
> Thank you for your thorough review, which improved our paper. We believe that some minor points can be better addressed during the discussion phase. We welcome any further questions and will respond promptly.
>
> References:
>
> [1] https://data.eumetsat.int/extended?query=MHS
>
> [2] arXiv:2407.11696
>
> [3] arXiv:2407.15586
>
> [4] MultiMAE (ECCV'2023)

---

> > ### Comment · Reviewer_Ht96 · 2025-08-03
> >
> > Thank you for the detailed response. I do not have any further questions.
> >
> > Just a remark: **Please do not include links in the rebuttal or discussion as this is against NeurIPS guideline**

---

> > > ### Author Response · Authors · 2025-08-04
> > > **Thanks for your kindly reply!**
> > >
> > > Dear Reviewer Ht96,
> > >
> > > We are very pleased to hear that all of your concerns have been addressed.
> > >
> > > To answer the Near-realtime Availability as Q7 required,  we provide reference [1] that is the official reference document from European Organisation for the Exploitation of Meteorological Satellites, a public satellite dataset that doesn't have any additional information about this paper.
> > >
> > > Please do not hesitate to raise any additional questions you may have, since we are now in the Author-Reviewer Discussion phase. We are committed to addressing all reviewer concerns and will ensure timely responses.  Thank you once again for your contributions to our work.

---

> > > > ### Comment · Reviewer_Ht96 · 2025-08-07
> > > > **Final justification**
> > > >
> > > > Thank you for the response. I still have an issue with how the baselines have been evaluated with this new protocol and I have decided to keep the rating.
> > > >
> > > > > To answer the Near-realtime Availability as Q7 required, we provide reference [1] that is the official reference document from European Organisation for the Exploitation of Meteorological Satellites, a public satellite dataset that doesn't have any additional information about this paper.
> > > >
> > > > NeurIPS guideline: **"Because of known concerns on identity leakage, we prohibit using any links in the rebuttal, including but not limited to anonymous or non-anonymous URL links, or updating your submitted github repository."**

---

> ### Author Response · Authors · 2025-08-08
> **Reply of evaluation setting**
>
> Dear Reviewer Ht96,
>
> Thank you very much for your valuable comments and for taking the time to review our work.
>
> To address your concerns comprehensively, we are more than happy to further clarify our evaluation methodology. In order to ensure that the test set covers a full year, we partitioned the data as follows: data from January 2012 to July 2021 was used for training, data from August 2021 to July 2022 for validation, and data from August 2022 to July 2023 for testing. The samples in our dataset are organized in 12-hour intervals. Thus, a lead time of 0–12h corresponds to a single-step prediction, 12–24h to a two-step prediction, and 24–36h to a three-step prediction. We report the mean absolute error (MAE) averaged over all observation sites for each of these lead time intervals.
> |Lead time: 0-12h |ATMS0|ATMS1|AMSU-A0(e-5)|AMSU-A1(e-5)|MHS0(e-4)|MHS1(e-4)|HIRS9|HIRS10|
> |-|-|-|-|-|-|-|-|-|
> |EarthNet|7.09±0.18|6.13±0.15|2.52±0.04|4.35±0.07|3.91±0.04|9.86±0.06|8.60±0.02|1.31±0.01|
> |Transformer-DOP|6.19±0.19|5.46±0.12|2.37±0.01|4.10±0.02|3.78±0.17|9.56±0.01|8.55±0.02|1.31±0.00|
> |Ours|**3.25±0.01**|**3.17±0.00**|**1.76±0.00**|**3.22±0.02**|**3.07±0.02**|**8.06±0.04**|**7.29±0.02**|**1.05±0.00**|
>
> |Lead time: 12-24h|ATMS0|ATMS1|AMSU-A0(e-5)|AMSU-A1(e-5)|MHS0(e-4)|MHS1(e-4)|HIRS9|HIRS10|
> |-|-|-|-|-|-|-|-|-|
> |EarthNet|11.29±0.43|9.06±0.09|4.07±0.09|6.56±0.08|5.35±0.02|13.42±0.06|10.31±0.05|1.85±0.03|
> |Transformer-DOP|10.02±0.37|7.99±0.19|3.65±0.02|5.86±0.05|5.23±0.06|12.94±0.08|10.20±0.05|1.78±0.01|
> |Ours|**7.97±0.34**|**6.59±0.19**|**3.08±0.07**|**5.04±0.09**|**4.48±0.08**|**10.94±0.06**|**9.07±0.04**|**1.48±0.01**|
>
> |Lead time: 24-36h|ATMS0|ATMS1|AMSU-A0(e-5)|AMSU-A1(e-5)|MHS0(e-4)|MHS1(e-4)|HIRS9|HIRS10|
> |-|-|-|-|-|-|-|-|-|
> |EarthNet|12.55±0.49|10.18±0.18|5.13±0.13|8.04±0.11|6.42±0.08|16.05±0.16|11.65±0.11|2.23±0.06|
> |Transformer-DOP|11.48±0.43|8.98±0.25|4.57±0.05|7.06±0.07|6.32±0.07|15.31±0.12|11.36±0.08|2.08±0.09|
> |Ours|**8.73±0.11**|**7.16±0.06**|**3.64±0.07**|**5.73±0.10**|**5.20±0.08**|**13.02±0.13**|**10.26±0.03**|**1.75±0.01**|
>
> While we cited the official reference document from the European Organisation for the Exploitation of Meteorological Satellites (EUMETSAT) to accurately indicate the near-real-time availability of the MHS data we used, we appreciate your kind reminder and sincerely apologize for any potential concerns this may have caused. We confirm that this citation does not involve any identity leakage.
>
> If you have any further questions or concerns, please feel free to raise them. We remain open and ready for further discussion.
>
> Best regards,
>  The Authors

---

### Official Review · Reviewer_HaE5 · 2025-07-02

**Clarity:** 3
**Significance:** 3
**Originality:** 2
**Rating:** 5
**Confidence:** 4

**Summary:**

The proposed data assimilation module based on a mask multi-modality autoencoder (MMAE) integrates multiple sources of observation data and transforms them into a uniform observation space. After data assimilation, it is treated as a standard spatiotemporal forecasting problem. In particular, in the forecasting stage, tokens from neighboring areas are padded to the tokens of the predicted region. In this way, the cross-regional boundary information of the border area is passed to the central forecasting region to make the forecasting smooth. Experiments are performed to demonstrate the good performance of the proposed method.

**Questions:**

1. As the data assimilation modular applies the multi-modal masked autoencoder, it can process different types of data, such as the observation data from weather stations. I would suggest the authors to expand it to weather station data.
2. Compare with more forecasting baselines.

**Ethical Concerns:**

["NO or VERY MINOR ethics concerns only"]

**Final Justification:**

The extensive experiments in the rebuttal include several baselines I have requested. And the experimental results clearly show the advange of the of the proposed method.

**Limitations:**

Yes.

**Quality:**

2

**Strengths And Weaknesses:**

Strengths:
1. The paper addresses an interesting problem and proposes a practical approach. By leveraging the observation data directly, the proposed method achieves good forecasting performance.
2. The data assimilation module based on multi-modal masked autoencoder sounds good. As the key contribution of this paper is the data assimilation modular, I would recommend highlight the general role of the data assimilation for different weather forecasting settings.
3. Experiments performed are well designed, such as visual comparison of cross-region boundary.

Weaknesses:
1. Technically, the forecasting stage is treated as a standard spatio-temporal forecasting problem. Except the cross-region boundary conditioning, the forecasting algorithm looks mediocre.
2. The compared baseline in forecasting is limited. Except Persistence (the naïve method which uses the last observation as the prediction), only EarthNet and Transformer-DOP are compared. It is encouraged to compare more baselines such as EarthFormer.

---

> ### Author Rebuttal · Authors · 2025-07-31
>
> We sincerely appreciate your constructive and detailed feedback. Thank you for your positive evaluation of our work’s novelty, methodological rigor, performance, and overall contribution. Your recognition that our paper addresses an interesting problem and proposes a practical approach is highly encouraging.
>
> ___
>
> > **Response to Q1: More Types of Data**
>
> Thank you for your helpful suggestion. We are currently exploring the possibility of extending DAWP to incorporate point-cloud MAE pretraining[1] combined with multi-modal MAE (MMAE)[2] for integrating weather station data. However, due to constraints in data collection, preprocessing, and ttraining ime, we were unable to include any experimental results in our rebuttal. We will add these experiments in the future version of our DAWP and include discussions into the revised manuscript.
>
> >  **Response to W2&Q2: Comparisons with More Baselines**
>
> Thank you for your suggestion. Due to length constraints in the main manuscript, we have moved the comparisons with additional baselines to the **Supplementary Material**. There, we include quantitative results comparing DAWP against six classical spatio-temporal forecasting models including RNN‑based, CNN‑based and Transformer‑based ones. We will ensure that future versions of our paper prominently present this baseline comparison in the main text, to better contextualize the strengths of our DAWP framework.
>
> | Lead time: 0-12h | AMSU-A-ch0 | AMSU-A-ch1 | ATMS-ch0 | ATMS-ch1 | HIRS-ch9 | HIRS-ch10 | MHS-ch0 | MHS-ch1 |
> |------------------|------------|------------|----------|----------|----------|-----------|---------|---------|
> | Persistence[3]      | 5.86       | 9.15       | 14.37    | 11.69    | 12.43    | 2.21      | 7.01    | 14.74   |
> | ConvLSTM[4]         | 127.82     | 208.9      | 73.21    | 78.44    | 77.4     | 11.05     | 62.27   | 158.57  |
> | PredRNN[5]          | 2.95       | 4.96       | 6.99     | 6.21     | 9.26     | 1.42      | 4.11    | 10.2    |
> | RainFormer[6]       | 3.95       | 6.52       | 9.08     | 8.05     | 10.41    | 1.63      | 4.98    | 11.69   |
> | EarthFormer[3]      | 18.97      | 33.94      | 37.33    | 38.75    | 22.49    | 3.62      | 18.00      | 55.04   |
> | SimVP[7]            | 3.61       | 6.02       | 7.29     | 6.61     | 9.84     | 1.53      | 4.68    | 11.35   |
> | TAU[8]              | 3.84       | 6.31       | 7.70      | 6.86     | 9.94     | 1.58      | 4.87    | 11.42   |
> | EarthNet[9]         | 2.93       | 4.89       | 6.96     | 6.14     | 9.15     | 1.39      | 3.96    | 9.65    |
> | Transformer-DOP[10]  | 2.67       | 4.48       | 6.40      | 5.61     | 9.22     | 1.42      | 3.91    | 9.48    |
> | **Ours**             | 1.92       | 3.39       | 3.36     | 3.27     | 7.7      | 1.12      | 3.07    | 7.91    |
>
> | Lead time: 12-24h | AMSU-A-ch0 | AMSU-A-ch1 | ATMS-ch0 | ATMS-ch1 | HIRS-ch9 | HIRS-ch10 | MHS-ch0 | MHS-ch1 |
> |-------------------|------------|------------|----------|----------|----------|-----------|---------|---------|
> | Persistence[3]       | 4.35       | 6.94       | 10.40     | 8.86     | 13.58    | 2.30       | 6.07    | 15.09   |
> | ConvLSTM[4]          | 128.13     | 211.14     | 81.22    | 82.6     | 71.64    | 9.98      | 59.26   | 145.44  |
> | PredRNN[5]           | 3.85       | 6.26       | 11.14    | 8.82     | 10.92    | 1.84      | 5.48    | 13.16   |
> | RainFormer[6]        | 11.46      | 15.72      | 19.43    | 17.76    | 18.75    | 3.28      | 11.74   | 32.10    |
> | EarthFormer[3]       | 18.98      | 33.96      | 37.33    | 38.76    | 22.50     | 3.62      | 18.01   | 55.03   |
> | SimVP[7]             | 4.83       | 7.74       | 11.48    | 9.10      | 11.49    | 2.02      | 6.43    | 14.2    |
> | TAU[8]               | 4.46       | 7.21       | 11.46    | 9.06     | 11.65    | 2.04      | 6.24    | 13.94   |
> | EarthNet[9]          | 4.12       | 6.65       | 11.25    | 9.00        | 11.14    | 1.98      | 5.46    | 13.11   |
> | Transformer-DOP[10]   | 3.84       | 6.14       | 10.04    | 8.04     | 11.08    | 1.95      | 5.19    | 12.65   |
> | **Ours**              | 3.11       | 5.12       | 7.35     | 6.35     | 9.57     | 1.54      | 4.51    | 10.54   |
>
> | Lead time: 24-36h | AMSU-A-ch0 | AMSU-A-ch1 | ATMS-ch0 | ATMS-ch1 | HIRS-ch9 | HIRS-ch10 | MHS-ch0 | MHS-ch1 |
> |-------------------|------------|------------|----------|----------|----------|-----------|---------|---------|
> | Persistence[3]       | 6.39       | 9.84       | 15.37    | 12.52    | 14.61    | 2.61      | 8.00       | 17.86   |
> | ConvLSTM[4]          | 128.42     | 211.83     | 82.03    | 85.17    | 72.62    | 10.15     | 60.44   | 148.62  |
> | PredRNN[5]           | 4.58       | 7.23       | 12.18    | 9.69     | 12.03    | 2.08      | 6.24    | 15.04   |
> | RainFormer[6]        | 24.89      | 32.34      | 35.49    | 36.54    | 30.78    | 4.92      | 21.51   | 69.68   |
> | EarthFormer[3]       | 18.98      | 33.96      | 37.34    | 38.77    | 22.51    | 3.63      | 18.02   | 55.04   |
> | SimVP[7]             | 5.72       | 8.91       | 12.93    | 10.33    | 12.71    | 2.32      | 73.95   | 15.99   |
> | TAU[8]               | 5.61       | 8.76       | 13.23    | 10.53    | 12.98    | 2.34      | 7.15    | 15.85   |
> | EarthNet[9]          | 5.17       | 8.14       | 12.52    | 10.08    | 12.37    | 2.36      | 6.41    | 15.08   |
> | Transformer-DOP[10]   | 4.91       | 7.54       | 11.35    | 9.07     | 12.39    | 2.27      | 6.22    | 14.70    |
> | **Ours**              | 3.66       | 5.80        | 7.84     | 6.81     | 10.71    | 1.79      | 5.15    | 12.22   |
>
> > **Response to W1: Design Choice of Forecasting Model**
>
> Thank you for recognizing our key contributions of a data assimilation stage and a forecasting stage for direct observation–based prediction, as highlighted in the review strengths. Although the forecasting model is simple, it still effectively supports the point that data assimilation initialization can aid direct observation forecasting. Moreover, our forecasting model efficiently resolve conflicts of input areas between regional data assimilation and global forecasting through the cross-region boundary conditioning.
>
> In addition, within the DAWP framework, the forecasting algorithm is flexible and has the potential to incorporate with more specialized forecasting methods in future work.
>
>
> > **Response to S2: Highlight DA's Role**
>
> Thank you for your valuable suggestion. The general role of Data Assimilation (DA) is to integrate observational data. Specifically, in physics-based numerical weather prediction systems, DA incorporates real-time observations into the model state to provide more accurate initialization for forecasting.
> In the context of Direct Observation-based Prediction, we leverage DA to fuse sparse observational inputs into dense, forecast-compatible representations. This helps mitigate key challenges such as the input–output mismatch, which often leads to inefficient roll-outs and unstable training dynamics. Following your suggestion, we will highlight the general role of DA more clearly in revised versions of the paper to better clarify its key contribution to our framework.
>
> ___
>
> We appreciate your feedback once again, which has helped us strengthen the overall presentation and completeness of the paper. If further clarification is needed, please do not hesitate to mention it, and we will promptly address your inquiries. We look forward to receiving your feedback.
>
> References:
>
> [1] Hess, Georg, et al. "Masked autoencoder for self-supervised pre-training on lidar point clouds." Proceedings of the IEEE/CVF winter conference on applications of computer vision. 2023.
>
> [2] Mizrahi, David, et al. "4m: Massively multimodal masked modeling." Advances in Neural Information Processing Systems 36 (2023): 58363-58408.
>
> [3] Gao, Zhihan, et al. "Earthformer: Exploring space-time transformers for earth system forecasting." Advances in Neural Information Processing Systems 35 (2022): 25390-25403.
>
> [4] Shi, Xingjian, et al. "Convolutional LSTM network: A machine learning approach for precipitation nowcasting." Advances in neural information processing systems 28 (2015).
>
> [5] Wang, Yunbo, et al. "Predrnn: A recurrent neural network for spatiotemporal predictive learning." IEEE Transactions on Pattern Analysis and Machine Intelligence 45.2 (2022): 2208-2225.
>
> [6] Bai, Cong, et al. "Rainformer: Features extraction balanced network for radar-based precipitation nowcasting." IEEE Geoscience and Remote Sensing Letters 19 (2022): 1-5.
>
> [7] Gao, Zhangyang, et al. "Simvp: Simpler yet better video prediction." Proceedings of the IEEE/CVF conference on computer vision and pattern recognition. 2022.
>
> [8] Tan, Cheng, et al. "Temporal attention unit: Towards efficient spatiotemporal predictive learning." Proceedings of the IEEE/CVF conference on computer vision and pattern recognition. 2023.
>
> [9] Vandal, Thomas J., et al. "Global atmospheric data assimilation with multi-modal masked autoencoders." arXiv preprint arXiv:2407.11696 (2024).
>
> [10] McNally, Anthony, et al. "Data driven weather forecasts trained and initialised directly from observations." arXiv preprint arXiv:2407.15586 (2024).

---

> ### Author Response · Authors · 2025-08-05
> **Looking forward to hearing from you**
>
> Dear Reviewer HaE5,
>
> Thank you once again for your thoughtful and constructive feedback.
>
> In response to your comments, we have added quantitative comparisons with six additional classical spatio-temporal baselines. We have also clarified the design choices for our forecasting model, including the role of Data Assimilation (DA) and the potential application of DAWP to station data.
>
> As we are now in the Author-Reviewer Discussion phase, please feel free to raise any further questions you may have. We are committed to addressing all reviewer concerns and ensuring timely responses. Thank you once again for your valuable contributions to our work.
>
> Best regards,
>
> The Authors of Submission 1085

---

> > ### Comment · Reviewer_HaE5 · 2025-08-06
> > **Thanks for the additional experiments.**
> >
> > 1. The extensive supplementary experiments look good to me. It shows the clear advantage of the proposed method. It is highly recommended to include it in the final manuscript.
> > 2. In terms of the weather station data, I understood the difficulty in such as short time, but still encourage the authors to explore it seriously in the future.

---

> > > ### Author Response · Authors · 2025-08-08
> > >
> > > Dear Reviewer HaE5,
> > >
> > > Thank you for your recognition and positive feedback. We are pleased that our revisions have adequately addressed your previous concerns. We will carefully revise the manuscript following your suggestions, including the supplemental experiments into the final version. We also sincerely appreciate that your understanding regarding the challenges of introducing weather station data in such as a short time. We will thoroughly explore the incorporation of the station in the future. If there are any further questions, feel free to raise them, and we are always ready for the discussion.
> > >
> > > Best regards,
> > >
> > > The Authors

---

### Author Response · Authors · 2025-08-08
**General response**

We sincerely thank the reviewers, Area Chairs (ACs), Senior Area Chairs (SACs) and Program Chairs (PCs) for their time, efforts, and thoughtful feedback throughout this review process.

We are especially grateful to the reviewers for their insightful suggestions, constructive comments, and encouraging evaluations.

---
We are encouraged that **our reviewers recognize this work**:

* **Reviewer** ```HaE5```:
    * "addresses an interesting problem and proposes a practical approach", "a good multi-modal masked autoencoder for data assimilation", "well-designed experiments".

* **Reviewer** ```Ht96```:
    * "addresses a critical problem in weather forecasting", "a clear framework presentation for readers".

* **Reviewer** ```RPch```:
    * "robust dataset and experimental setup that addresses the interests and questions of both the Climate AI and ML/CV communities", "effectively leverages multiple modalities or sensors through both a timely and appropriate approach".

* **Reviewer** ```bPKH```:
    * "a novel framework addressing the limitations of existing methods", "the contribution to the field of weather prediction is not incremental but rather innovative".

---

In response to your valuable feedback, we have undertaken the following clarifications and improvements:

* Experiments:

    * As suggested by **Reviewer** ```HaE5```, we will move a comprehensive comparison of observation-based forecasting methods and classical spatiotemporal baselines from the supplementary materials into the main manuscript, showing the clear advantage of our method.

    * As suggested by **Reviewer** ```Ht96```, we expanded our experiments with multiple random seeds and showed that our improvements are statistically significant.

    * As suggested by **Reviewer** ```Ht96```, we added evaluation results on a full year’s test data, further ensuring a representative evaluation.

    * As suggested by **Reviewer** ```Ht96```, we included computational resource requirements for DAWP, including inference time, number of parameters, and memory usage. Key hyperparameters for all baselines are now reported.

    * As suggested by **Reviewer** ```RPch```, we added the Probability of Detection (POD) metric to the precipitation prediction evaluation, providing a multi-faceted assessment.

* Elaboration and Clarity:
    * As suggested by **Reviewer** ```HaE5```, we have discussed the design of our forecasting model for adapting the data assimilation module and highlighted the impacts of data assimilation in different settings.

    * As suggested by **Reviewer** ```Ht96```, we discussed distinctions between direct observation-based models and reanalysis-based models, including their data sources and architectural implications, to better clarify the gap between observation and physical space.

    * As suggested by **Reviewer** ```RPch```, we improved the methodological clarity by discussing how LT3P, ClimODE, and VideoMAE relate to and differ from our approach. We have clarified our core contribution—identifying the challenge of observation-based forecasting due to gaps between sparse input and dense prediction—and addressed it through our novel DAWP framework.

    * As suggested by **Reviewer** ```bPKH```, we clarified how the data assimilation stage enhances trustworthiness, and how modality ablation provides post-hoc explainability. We also discussed the potential integration with PINNs for further improvements in both aspects.

* Other Minor Modifications:
    * As suggested by **Reviewer** ```HaE5```,  we mentioned the future incorporation of station data in our work.

    * As suggested by **Reviewer** ```Ht96```, we clarified the reason behind our dataset’s 35TB scale, as well as the naming choices for abbreviations like Transformer-DOP and AIWP.

    * As suggested by **Reviewer** ```Ht96```, we added a reference document to demonstrate the near-real-time availability of our satellite data.

    * As suggested by **Reviewer** ```Ht96```, we explained how the choices of window length and sub-image size are guided by satellite revisit cycles and sequence lengths.

    * As suggested by **Reviewer** ```RPch```, we expanded the related work section with additional references, such as LT3P and OMG-HD, to provide broader context.

For detailed responses regarding each of the above aspects, please kindly refer to the response rebuttal windows in the review section.

---
Last but not least, we sincerely thank our reviewers, ACs, and PCs again for the valuable time and efforts devoted and the constructive suggestions provided during this review.

---
Yours sincerely,

The Authors

---

### Note · Authors · 2025-08-14

We sincerely thank all reviewers for your thoughtful engagement and constructive feedback.

---

We are encouraged by the consistent recognition of our approach, DAWP, as a novel and technically sound framework for observation-based forecasts, particularly in the context of satellite-based observation forecasts. The positive feedback on our well-designed and robust dataset, experiments, and our not incremental but rather innovative contribution to the field of weather prediction has been invaluable.

---

During the discussion phase, we addressed all raised concerns with concrete clarifications and updates:

**Experiments**: We clarified the comparisons between six classical spatio-temporal prediction methods and our DAWP, emphasizing the need for observation-based forecasts to tackle the sparsity issue. We also re-conducted experiments with four random seeds and tested over a full year to show statistically significant improvements of DAWP and MASK-VIT-VAE. Additionally, we benchmarked the resource usage of each component of DAWP and included the Probability of Detection (POD) metric to thoroughly evaluate the downstream task of precipitation prediction, demonstrating DAWP’s consistent advantage across various metrics.

**Relation to Prior Work**: Our method, based entirely on satellite observations, addresses the global forecasting challenge in the absence of reanalysis data. We introduced a unique strategy that combines regional observation completion with cross-region forecasting, resolving the observation-forecast mismatch seen in prior methods during rollout. This significantly enhances forecasting accuracy over longer time horizons.

**Extensions**: We have outlined our plan to adapt the method for weather station data, showcasing DAWP’s potential to generalize across heterogeneous data sources. Furthermore, we discussed the explainability and trustworthiness of observation-forecast methods, emphasizing the importance of transparency in model predictions.

These clarifications reinforce the novelty, robustness, and scalability of DAWP.

---

By addressing the challenges of observation-based forecasting, our work enables **more accurate observation predictions**, providing a strong foundation for future research and practical applications.

---

Once again, we sincerely thank you for the time and effort you devoted to reviewing our submission!

Yours sincerely,

The Authors of Submission 1085

---

### Decision · Program_Chairs · 2025-09-17

**Decision:**

Accept (poster)

**Comment:**

This paper introduces DAWP, a two-stage framework (AIDA initialization followed by AIWP forecasting) for global observation forecasting in satellite observation space. The approach moves beyond reliance on reanalysis data, addressing the challenge of learning spatiotemporal dynamics from irregular, high-resolution satellite observations. The core contribution is the mitigation of the mismatch between sparse observational inputs and dense prediction outputs. Extensive experiments benchmark against multiple classical spatio-temporal prediction baselines and demonstrate improved performance, including in global precipitation forecasting tasks.

The work tackles an important and timely problem by enabling direct forecasting in observation space without analysis data dependency. The experiments are decently strong and there is broad spectrum of baselines. The authors addressed specific requests for more baseline comparisons, more detail on experimental protocols, full-year test splits etc. There are still concerns w.r.t. how the baselines have been adapted to this setting as the protocol and settings are different. Would be good to further emphasize differences w.r.t. EarthNet and Transformer-DOP lines of work in the observation based forecasting setting.

Overall the paper is an ambitious attempt to advance direct observation forecasting. Despite some concerns w.r.t baselines, the work is likely to stimulate further research. I encourage the authors make sure the results are reproducible by the community, including the baselines.